# Subgroups Matter for Robust Bias Mitigation

**Anissa Alloula** [1]   **Charles Jones** [2]   **Ben Glocker** [2]   **Bartłomiej W. Papież** [1]

## Abstract

Despite the constant development of new bias mitigation methods for machine learning, no method consistently succeeds, and a fundamental question remains unanswered: when and why do bias mitigation techniques fail? In this paper, we hypothesise that a key factor may be the often-overlooked but crucial step shared by many bias mitigation methods: the definition of subgroups. To investigate this, we conduct a comprehensive evaluation of state-of-the-art bias mitigation methods across multiple vision and language classification tasks, systematically varying subgroup definitions, including coarse, fine-grained, intersectional, and noisy subgroups. Our results reveal that subgroup choice significantly impacts performance, with certain groupings paradoxically leading to worse outcomes than no mitigation at all. Our findings suggest that observing a disparity between a set of subgroups is not a sufficient reason to use those subgroups for mitigation. Through theoretical analysis, we explain these phenomena and uncover a counter-intuitive insight that, in some cases, improving fairness with respect to a particular set of subgroups is best achieved by using a different set of subgroups for mitigation. Our work highlights the importance of careful subgroup definition in bias mitigation and presents it as an alternative lever for improving the robustness and fairness of machine learning models.

## 1. Introduction

A significant barrier to the wider deployment of machine learning (ML) models is their tendency to fail when tested on distributions that differ from their training data. One particularly concerning manifestation of this issue is performance degradation for population subgroups, often caused

[1]University of Oxford, UK [2]Imperial College London, UK. Correspondence to: Anissa Alloula <anissa.alloula@dtc.ox.ac.uk>.

*Proceedings of the 42$^{nd}$ International Conference on Machine Learning*, Vancouver, Canada. PMLR 267, 2025. Copyright 2025 by the author(s).

by bias in training data such as spurious correlations (SC), under-representation of certain subgroups, or shifts in the presentation of the target $Y$ (Jones et al., 2024). Bias mitigation methods aim to address these issues by training more robust models which are less susceptible to these biases, thereby improving generalisation. These methods generally adapt model training to improve the performance of some disadvantaged subgroups within the training data.

Despite the number of bias mitigation methods which have been proposed, benchmarks are increasingly reporting that the performance of these methods is inconsistent when tested in new settings, and that they often fail to surpass the empirical risk minimisation (ERM) baseline (Zong et al., 2023; Zietlow et al., 2022; Chen et al., 2023; Shrestha et al., 2022; Alloula et al., 2024). Some efforts have been made to begin to elucidate the conditions under which certain bias mitigation methods might be valid, such as the work of Jones et al. (2025) in fair representation learning and Schrouff et al. (2024) in data balancing. However, the choice of an appropriate mitigation method is only one aspect of the problem. To successfully mitigate bias, we must also select which subgroups we wish to apply the methods on – a question which very little work has explicitly addressed.

Indeed, most bias mitigation methods rely on some form of grouping to first identify disadvantaged subgroups within the training data and then to implement group-based strategies aimed at improving generalisation or fairness. This can be as simple as observing a disparity in model performance between men and women, and trying to fix this by rebalancing the training data such that both subgroups are more uniformly distributed (Weng et al., 2023), or noticing that a model performs poorly on data coming from a specific type of scanner, and applying a robust learning strategy such as adversarial training to prevent learning of scanner-specific but task-irrelevant information (Ganin & Lempitsky, 2015). To date, the literature has predominantly focused on simple and coarse subgroups, for example blonde hair and non-blonde hair in the CelebA dataset (Liu et al., 2015b) or waterbirds and landbirds in the waterbird dataset (Sagawa* et al., 2020). In medical applications, the subgroups are often "white" or "non-white", "men" or "women", or some coarse binning of age (Ricci Lara et al., 2022). Subgroup choice is often motivated by two factors: a) practical constraints, for instance only having annotations for common

attributes, and b) ethical or societal goals to achieve fairness with respect to specific subgroups. However these subgroups may poorly capture the underlying cause of model underperformance, thus obscuring critical information for bias mitigation methods.

In this work, we aim to better understand whether we can optimise this crucial step of subgroup definition in the same way that new bias mitigation methods are optimised. We investigate the role of subgroup definition on the performance of these methods, and whether poor subgroup definition might explain why these methods often fail. We construct a setting of bias inspired by a real-world chest radiograph example (Olesen et al., 2024) where there is a spurious correlation during training but which is absent during testing, and which is present in different proportions across subgroups, resulting in disparities in model performance. We consider realistic ways in which subgroups might be generated based on relevant attributes, and explore how they impact the performance of four state-of-the-art bias mitigation methods in four semi-synthetic vision and language datasets. We identify key patterns in the performance of these methods across groupings. Certain groupings lead to a large improvement over the ERM baseline, while others substantially lower performance relative to the baseline. We propose that the effectiveness of a given subgrouping strategy is linked to its ability to recover the unbiased test distribution. We summarise the key contributions of this work as follows:

- We show that the groupings used for bias mitigation strongly affect how well each method works, and provide insights on optimal grouping strategies.

- We argue that observing a disparity in model performance across a set of subgroups does not justify using those subgroups for mitigation, and may in fact, make matters worse.

- We provide a possible explanation for the differences in subgroup effectiveness based on the minimum KL divergence between the subgroup-weighted biased distribution and the unbiased test distribution.

- We challenge the conventional assumption that the best way to obtain "fairness" with respect to a specific set of subgroups is always achieved by using those same subgroups for bias mitigation.

## 2. Related work

### 2.1. Bias identification

Research on bias detection has increasingly focused on refining subgroup definitions to capture complex patterns of unfairness. While individual fairness, as introduced by Dwork et al. (2012), offers a theoretically elegant approach by evaluating fairness at the individual level, its practical challenges have limited its adoption. As a result, group-based analyses remain the dominant paradigm. More recently, efforts have been made to move beyond traditional binary categories (e.g., men/women or white/non-white) to identify disparities that such coarse classifications may obscure. For instance, Kearns et al. (2018) and Buolamwini & Gebru (2018) illustrate how failing to account for intersections of attributes can entirely hide performance disparities. Xu et al. (2024) discuss the difficulty of identifying intersectional bias when there are a number of attributes at play, and propose a generative approach to discover high-bias intersections amongst many possibilities. Similarly, Movva et al. (2023) demonstrate that in clinical risk prediction models, variation in performance *within* 4 commonly-used ethnic groups often exceeds the variation *between* these coarse groups, advocating for the use of much more precise categories to describe ethnicity. These works highlight how heterogenous subgroups on which unfairness is observed can be, yet most of them do not consider how this translates to conducting bias mitigation.

### 2.2. Subgroup definition for bias mitigation

A smaller body of work has considered how this problem of subgroup definition extends to bias mitigation. For example, Awasthi et al. (2020), Wang et al. (2020a), and Stromberg et al. (2024) explore how noise in subgroup annotations may impact post-processing, distributionally robust optimisation (DRO), and last-layer-retraining respectively. They find that fairness is not guaranteed under some perturbations of the true attributes. Ghosal & Li (2023) also extend group DRO (gDRO) with probabilistic subgroup labels if there is uncertainty with respect to the subgroup annotations. Li et al. (2023); Kim et al. (2024) show the difficulty of mitigating bias when there are multiple spurious correlations and annotated subgroups. Perhaps the work most closely related to ours is Zhou et al. (2021), which considers a setting where a spurious correlation is causing bias, and show that gDRO fails in an example where the subgroups do not directly account for the spurious correlation. They attribute this failure to the inability to upweight bias-conflicting samples effectively, as their constructed "imperfect" groups also include spuriously correlated samples - though their subgroup construction appears somewhat unrealistic. All of these works point to a sensitivity of bias mitigation methods to the subgroups defined, however they each only explore one possible flaw in subgroup annotations and restrict their scope to a single bias mitigation method.

### 2.3. Bias mitigation without subgroups

While the above works explore some possible failures in subgroup definition for common bias mitigation methods,

others have developed new methods altogether which do not require subgroups to be defined in the traditional way. For instance, Kearns et al. (2018) and Hebert-Johnson et al. (2018) propose algorithms which aim to achieve fairness across all identifiable or richly structured subgroup classes. Bias discovery methods bypass the question of pre-defining subgroups altogether, and are useful in settings where subgroup annotations are missing. Some methods decouple bias discovery from bias mitigation by first "discovering" biases through inferring subgroup annotations with an external model (Han & Zou, 2024; Marani et al., 2024) or by clustering points based on their feature representations (Krishnakumar et al., 2021), and subsequently performing bias mitigation. Other methods forgo explicit subgroup identification and instead rely on some form of regularisation or upweighting of misclassified samples (Ahn et al., 2022; Park et al., 2024; Liu et al., 2021).

We restrict our main analysis to established mitigation methods which use subgroup annotations because (a), they often serve as the upper bound for mitigation methods (Zhou et al., 2021; Pezeshki et al., 2021; Bayasi et al., 2024), (b), many nominally label-free methods still require subgroup annotations for validation or hyperparameter tuning (Pezeshki et al., 2024), and (c), label-free methods frequently infer subgroups, making it still essential to understand their role.

## 3. Background

### 3.1. Overview of bias mitigation methods

#### 3.1.1. EMPIRICAL RISK MINIMISATION

Traditional deep learning models conduct empirical risk minimisation (**ERM**), where, for given inputs $x \in \mathcal{X}$ and labels $y \in \mathcal{Y}$ from a distribution $\mathcal{P}$, the objective is to find a model $\theta \in \Theta$, that minimises the expected loss $\mathbb{E}_{\mathcal{P}}[\ell(\theta; (x, y))]$. However, in practice, only a subset of $\mathcal{P}$, denoted $\mathcal{P}_{train}$, is available, so the loss over all samples in $\mathcal{P}_{train}$ is minimised:

$$\hat{\theta}_{\text{ERM}} := \arg\min_{\theta \in \Theta} \mathbb{E}_{(x,y) \sim \mathcal{P}_{train}}[\ell(\theta; (x, y))]. \quad (1)$$

#### 3.1.2. COMMON BIAS MITIGATION METHODS

Minimising the average loss over $\mathcal{P}_{train}$ often results in poor generalisation and poor performance on minority subgroups in the data. To address these shortcomings, bias mitigation methods have been proposed as alternatives to ERM. In this work, we focus on 4 commonly used bias mitigation methods which have demonstrated state-of-the-art results in certain tasks and represent the broad spectrum of existing methods. We define $k$ disjoint subgroups $\mathcal{P}_g$ indexed by $\mathcal{G} = \{1, ..., k\}$, which partition $\mathcal{P}_{train}$. We assume each training sample is annotated with its subgroup label $g$, thus giving the tuples $(x, y, g)$; however, subgroup

information may be unavailable at inference time.

**Group distributionally robust optimisation (gDRO)** (Sagawa* et al., 2020) reweights samples during loss calculation. The objective is to minimise the worst-case expected loss across each subgroup $\mathcal{P}_g$, (Equation (2)). In practice, an online optimisation algorithm is used to assign higher weight to high-loss subgroups in each batch.

$$\hat{\theta}_{\text{gDRO}} := \arg\min_{\theta \in \Theta} \left\{ \max_{g \in \mathcal{G}} \mathbb{E}_{(x,y) \sim \mathcal{P}_{train_g}}[\ell(\theta; (x, y))] \right\}. \quad (2)$$

**Resampling** (Idrissi et al., 2022) also relies on reweighting but achieves it by adjusting the sampling probability of each subgroup $\mathcal{P}_g$ at the batch level so that each batch is balanced across subgroups. The loss for resampling is equivalent to:

$$\hat{\theta}_{\text{resampling}} := \arg\min_{\theta \in \Theta} \sum_{g=0}^{k} \frac{1}{k} \mathbb{E}_{(x,y) \sim \mathcal{P}_{train_g}}[\ell(\theta; (x, y))]. \quad (3)$$

**Domain Independent (DomainInd)** learning adjusts the model architecture by replacing the single classifier head with $k$ separate classifier heads, each corresponding to a subgroup, such that although each sample is passed through the same encoder, the decoder can be fine-tuned to each subgroup (Wang et al., 2020b). At inference, the classifier head with the largest activation makes the final prediction.

**Conditional learning of fair representations (CFair)** aims to learn fair and robust representations of the target label independent of any subgroup information (Zhao et al., 2020). This is achieved by aligning conditional representations of samples from different subgroups.

#### 3.1.3. SUBGROUP DEFINITION ACROSS THESE METHODS

In this work, we distinguish between two categories of bias mitigation methods: reweighting (gDRO and resampling), and model-based methods (DomainInd and CFair). This is because of differences in how subgroups are generally defined. While in model-based methods subgroups are defined solely based on attribute(s) $A$, reweighting methods can additionally define subgroups over $\mathcal{A} \times \mathcal{Y}$. This is because the former methods perform best if each subgroup contains both positive and negative samples. We expand on this distinction and its implications in Appendix A.2.

### 3.2. Problem setting

We frame the task according to the fairness paradigm described in Jones et al. (2025), whereby the objective is to generalise from a biased training distribution to an unbiased

*Figure 1.* Causal graphs representing interactions between $Y$, $A$, and $S$ variables in the training data and in the unbiased test data. Conditioning on selection in the training data results in spurious correlations between $Y$, $A$, and $S$. Coloured bars also illustrate the proportions of $Y$, $A$, and $S$ combinations in both settings.

testing distribution. Dataset bias can take many forms (as discussed in Jones et al. (2024)), but here, we focus on bias arising from spurious correlations between certain attributes of the data and the class label $Y$ which disappear in the unbiased deployment setting. We define two binary attributes $A$ and $S$ whose information is encoded in $X$ in the form of latent features $X_A$ and $X_S$ respectively. Features relating to the true class $Y$ are represented as $X_Y$. In the unbiased test setting, attribute-related information is independent of $Y$, such that $P(Y) = P(Y \mid A) = P(Y \mid S)$. However, in $P_{train}$, mechanisms like data selection may lead to the violation of this independence. Causal graphs illustrating this scenario are shown in Figure 1.

In particular, in this study, we consider an example of a spurious correlation between $A$ and $Y$. Furthermore, we consider that the spuriously correlated samples are unevenly distributed across the second attribute $S$. This is inspired from a real world medical imaging example from Olesen et al. (2024). Specifically, they describe a chest X-ray diagnosis model which shows better performance in one sex subgroup (denoted here by $S$). Previous attempts to reduce this disparity, such as resampling of the data across $S$, and manipulation of sex-specific regions of the X-rays, were ineffective, leading to the hypothesis that sex-specific differences were not causing the disparity (Weng et al., 2023). Subsequent work revealed that men and women presented different proportions of chest drains and ECG wires, which the model used as spurious correlations to predict disease, and that, balancing the test data with respect to these artefacts (corresponding to $A$ in this work) resulted in equal performance across sexes. To mimic this example, we use a semi-synthetic setting where the subgroup $S = 0$ contains 95% spuriously correlated samples, while the subgroup $S = 1$ contains a smaller, though still substantial, proportion of spuriously correlated samples (80%). With this setup, there would be a substantial difference in a model's performance across subgroups if it only correctly classified

samples satisfying this correlation.

We represent each distribution as probability vectors in $\mathbb{R}^8$ with each element corresponding to the probability of sampling one $(Y, S, A)$ subgroup. The unbiased distribution is defined to be uniform, i.e., $\mathcal{P}_{\text{unbiased}} = \left[\frac{1}{8}, \frac{1}{8}, \dots, \frac{1}{8}\right]$, while $\mathcal{P}_{\text{train}} = \left[\frac{0.95}{4}, \frac{0.05}{4}, \frac{0.8}{4}, \frac{0.2}{4}, \frac{0.05}{4}, \frac{0.95}{4}, \frac{0.2}{4}, \frac{0.8}{4}\right]$. Further details are presented in Table 1 and Figure 1.

*Table 1.* Probability distributions across $Y$, $A$, and $S$ in the biased train and validation dataset and the unbiased test set.

| Probability distributions | $\mathcal{P}_{\text{train}}$ | $\mathcal{P}_{\text{unbiased}}$ |
|---|---|---|
| $P(Y = 1)$ | 0.5 | 0.5 |
| $P(A = 1)$ | 0.5 | 0.5 |
| $P(Y = 0 \mid S = 0) = P(Y = 0 \mid S = 1)$ | 0.5 | 0.5 |
| $P(Y = 0 \mid A = 0) = P(Y = 1 \mid A = 1)$ | 0.875 | 0.5 |
| $P(Y = 0, A = 0 \mid S = 0) = P(Y = 1, A = 1 \mid S = 0)$ | 0.95 | 0.5 |
| $P(Y = 0, A = 0 \mid S = 1) = P(Y = 1, A = 1 \mid S = 1)$ | 0.8 | 0.5 |

## 4. Experiments

### 4.1. Subgroup generation

To understand how different subgroup definitions affect the generalisation performance of bias mitigation methods, we construct multiple sets of subgroups based on $A$, $S$, and $Y$ and train each model with them. Our goal is to simulate realistic scenarios, for instance where one may only have access to certain variables or to noisy subgroup annotations, or when deciding between the use of coarse or fine-grained subgroups (e.g., a level of ethnicity categorisation, or discretising on a continuous variable like age). We denote subgroups constructed as the intersection of multiple variables $a \times b$ as $(a, b)$.

For data-based methods, these include: 1) subgroups based on a single variable: $A$, $Y$, and $S$ 2) subgroups based on the intersection of two or three variables: $(A, Y)$, $(S, Y)$, and

$(Y, S, A)$ 3) SC/no-SC subgroups where we group the two bias-aligned $(A, Y)$ subgroups together and the two bias-conflicting $(A, Y)$ subgroups together 4) subgroups based on the random splitting of existing subgroups: $(A, Y)_8$, $(S, Y)_8$ which split each $(A, Y)$ and $(S, Y)$ subgroup in two such that there are 8 subgroups in total 5) 4 completely random subgroups. For model-based methods, we do not include subgroupings based on $Y$ and also add $A_4$, $S_4$ ($A$ and $S$ subgroups randomly split in two respectively), and $(A, S)$ subgroups. Finally, we explore 6) the impact of noise in subgroup annotations by injecting noise (in the form of mislabelling) into 1 - 50% of the $(A, Y)$ and $A$ subgroup annotations for data- and model-based methods respectively. This noise does not affect the class labels $Y$. For Civil_comments, in addition to the synthetic granular subgroups, we directly explore the impact of granularity on real subgroups, as the dataset contains subgroup information of multiple hierarchy levels.

In total, we consider 15 subgroup combinations for reweighting-based methods and 12 for model-based methods. We illustrate these subgroups in Figure A6. This comprehensive set of subgroup definitions allows us to systematically investigate the impact of subgroup choices on the effectiveness of bias mitigation methods.

### 4.2. Datasets and tasks

We evaluate performance in image classification tasks in four datasets which we construct to satisfy the distributions specified in Figure 1. We summarise all details in Table A5.

We adapt the **MNIST** dataset (Lecun et al., 1998) by binarising the classification task into predicting whether a digit is even or odd ($Y$). We add additional attributes by modifying the image background colour ($A$) to black or white, and colouring the foreground ($S$) as red or green. This controlled setting allows for clear evaluation of subgroup influences in a simple task.

To explore a more challenging and realistic setting, we repeat the experiments with chest X-ray images from the CheXpert dataset (**CXP**) (Irvin et al., 2019). The task is classification of the presence of pleural effusion ($Y$). $S$ is the sex of the patient and $A$ the presence of a pacemaker, using annotations provided in (Anthony & Kamnitsas, 2023).

We explore another real vision dataset commonly used in fair ML research, **CelebA** (Liu et al., 2015a). The task is binary classification of whether the individual has blonde hair ($Y$) with additional attributes $A$, perceived gender, and $S$, whether the individual is smiling.

Finally, we explore whether our findings extend to the text modality through the use of the **Civil_comments** dataset, also commonly used in fair ML (Borkan et al., 2019). The target $Y$ is toxicity prediction, with $A$ being any mention of

gender, and $S$ any mention of religion. Additionally, as this dataset contains multiple levels of subgroup annotations, we do a real experiment on the impact of subgroup granularity (instead of simply randomly splitting subgroups in two). We compare mitigation on the $A$ groups to mitigation on granular $A$ groups (e.g., any mention of males, any mention of another gender, and no mention of any gender), and likewise for $S$ (e.g., any mention of the Christian religion specifically, any mention of a non-Christian religion, and no mention of religion).

### 4.3. Implementation details

We implement and train models with each of the gDRO, resampling, DomainInd, and CFair bias mitigation methods. We apply each method to each of our generated subgroups and average the results over three random seeds. We repeat this process for the four datasets, comparing performance of the bias mitigation methods with the baseline ERM method. In total, we train 306 models, with $\sim$40 NVIDIA A100 hours of compute. The training strategy, hyperparameters, architectures etc. are the same across all models, as detailed in Table A7, except for necessary adjustments to apply each bias mitigation method. The code is available here.

We report the mean and standard deviation of the aggregate area under receiver operating characteristic curve (AUC) on the unbiased test set (following the fairness paradigm described in Jones et al. (2025)), alongside worst-group accuracy and accuracy gap across subgroups. We select these measures for their directness and simplicity compared to other fairness criteria. We do *not* vary subgroup definition for evaluation and only evaluate accuracy with respect to $S$ and $A$ subgroups, as we are simply interested in how subgroup definition affects the mitigation process.

## 5. Results and discussion

### 5.1. ERM performance drops on the unbiased test set

The baseline model, trained with no bias mitigation, shows a sharp drop in performance when tested on the unbiased test set, with a decrease of 0.10 to 0.25 in AUC for all datasets (Table 2). This drop occurs because the model has learned to rely on the spurious attribute $A$ as a proxy for $Y$, and this correlation is absent in the unbiased test set. Moreover, the ERM model exhibits disparities in performance. This is particularly pronounced for the $S$ subgroups, both on the biased validation set and on the unbiased test set. A standard approach would therefore have been to apply mitigation to the $S$ or $(S, Y)$ subgroups. In the following sections, we explore whether various bias mitigation methods, used with different groupings, can improve test set performance and reduce these disparities.

*Table 2.* The baseline model performance shows a sharp decrease on the unbiased test set for all datasets. Subgroup-wise accuracy also reveals large disparities with respect to $S$ in both datasets. Mean and standard deviation across three random seeds are shown.

| Accuracy | MNIST | | CXP | | Civil_comments | | CelebA | |
|---|---|---|---|---|---|---|---|---|
| | Val | Test | Val | Test | Val | Test | Val | Test |
| **Overall** | $0.943 \pm 0.012$ | $0.698 \pm 0.054$ | $0.898 \pm 0.007$ | $0.659 \pm 0.007$ | $0.886 \pm 0.003$ | $0.726 \pm 0.024$ | $0.954 \pm 0.003$ | $0.865 \pm 0.007$ |
| **Min** ($A$) | $0.936 \pm 0.025$ | $0.694 \pm 0.003$ | $0.869 \pm 0.013$ | $0.554 \pm 0.020$ | $0.878 \pm 0.006$ | $0.724 \pm 0.037$ | $0.952 \pm 0.002$ | $0.838 \pm 0.019$ |
| **Gap** ($A$) | $0.014 \pm 0.026$ | $0.008 \pm 0.112$ | $0.058 \pm 0.014$ | $0.220 \pm 0.028$ | $0.016 \pm 0.013$ | $0.004 \pm 0.047$ | $0.005 \pm 0.007$ | $0.055 \pm 0.025$ |
| **Min** ($S$) | $0.917 \pm 0.024$ | $0.599 \pm 0.061$ | $0.841 \pm 0.010$ | $0.633 \pm 0.020$ | $0.837 \pm 0.002$ | $0.726 \pm 0.029$ | $0.940 \pm 0.004$ | $0.861 \pm 0.002$ |
| **Gap** ($S$) | $0.052 \pm 0.025$ | $0.194 \pm 0.080$ | $0.116 \pm 0.015$ | $0.052 \pm 0.026$ | $0.095 \pm 0.009$ | $0.006 \pm 0.047$ | $0.029 \pm 0.008$ | $0.009 \pm 0.014$ |

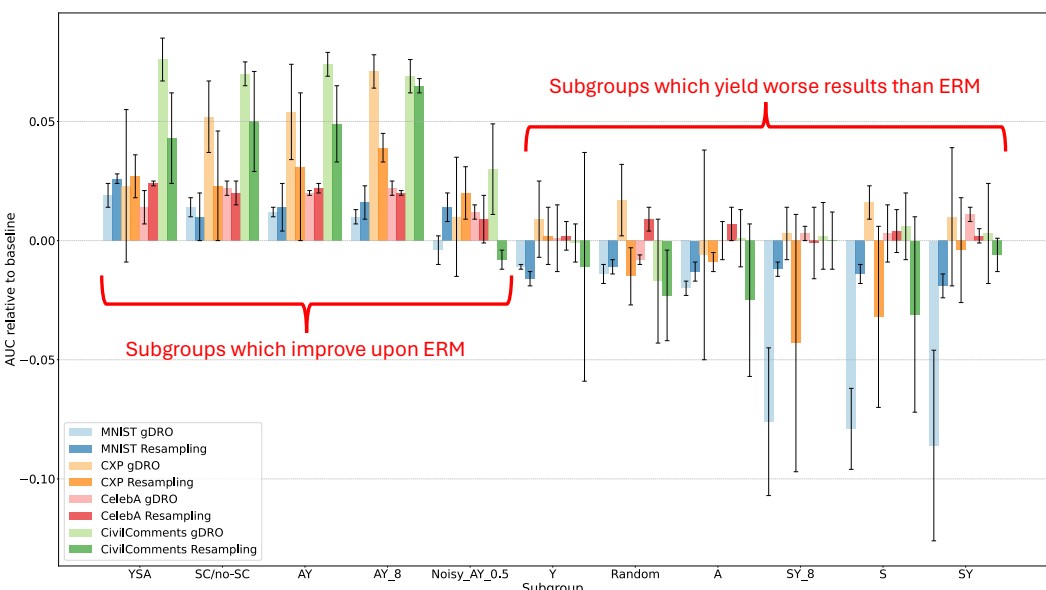

*Figure 2.* Performance on the $\mathcal{P}_{unbiased}$ in gDRO and resampling is highly dependent on the subgroups used across all four datasets. Bars represent overall change in AUC relative to the ERM baseline, with error bars indicating the standard deviation across 3 random seeds.

## 5.2. Groupings used for mitigation strongly impact bias mitigation performance

We find that test set performance is highly dependent on the subgroups used for mitigation, with some subgroups boosting overall performance by more than 0.07, while others reduce it by up to 0.08 (Figures 2 and A7). We first note that the four datasets reveal similar patterns across groupings for reweighting-based bias mitigation methods and model-based methods, suggesting that there are universal trends relating to the bias setting which generalise outside of the specific dataset, task, and mitigation method. For reweighting-based methods, subgroups based on $(A, Y)$, i.e., $(A, Y)$, $(A, Y)_8$, $(Y, S, A)$, $(A, Y)$ with small levels of noise, and SC/no-SC subgroups, enable the model to focus on the $(A, Y)$ pairs without the spurious correlation during training. Therefore the model learns to predict $Y$ independently of $A$, leading to better generalisation performance. Conversely, subgroups which do not take $(A, Y)$ information into account tend to result in worse performance than the baseline model (for instance $S$, $(S, Y)$, $(S, Y)_8$, $Y$, $A$, and Random subgroups), as they fail to guide the model

away from relying on the spurious attribute $A$. For model-based methods, a similar pattern is evident (Figure B7); $A$ subgroups generally present the best performance (analogous to $(A, Y)$ in data-based methods). For instance, with $A$ subgroups in DomainInd, performance is increased by 0.07 in the CXP dataset. On the other hand, $S$ subgroups consistently harm performance, decreasing it by up to 0.14 for CFair in MNIST.

**Applying bias mitigation with certain groupings can lead to worse outcomes than ERM.** For instance, $S$ subgroups are detrimental to generalisation performance in 10 out of 16 experiments, and have no effect in the remaining 4 (Figures 2, A7). They are detrimental despite the fact that there is a substantial disparity in performance between them (Table 2). This observation suggests that in the absence of information about the underlying mechanism causing bias, it may be better to refrain from using mitigation methods altogether, even for simple approaches like data balancing. We believe that this finding may partly explain the failures of bias mitigation methods reported in recent studies (Zong

et al., 2023; Zietlow et al., 2022; Chen et al., 2023; Shrestha et al., 2022), where inadequate subgroup definitions may have caused counter-productive bias mitigation.

**Increasing subgroup granularity has no significant effect on performance.** For example, there is little difference between results for $(A, Y)$ vs $(A, Y)_8$ and $(S, Y)$ vs $(S, Y)_8$ (Figure 2). This is most likely because the bias mitigation methods tested here are designed to maintain stability if there are many subgroups. In practice, this suggests that dividing subgroups into finer subgroups is unlikely to harm performance, and may even be beneficial if one is unsure which specific attributes may be responsible for underperformance. This insight could be relevant in settings where there are multiple levels of granularity to choose from (such as continuous data e.g., age, or multi-level categories e.g., ethnicity). We validate this with the real coarse and granular subgroups in Civil_comments, and indeed find no significant difference in unbiased generalisation for resampling and gDRO, as shown in Figure 3.

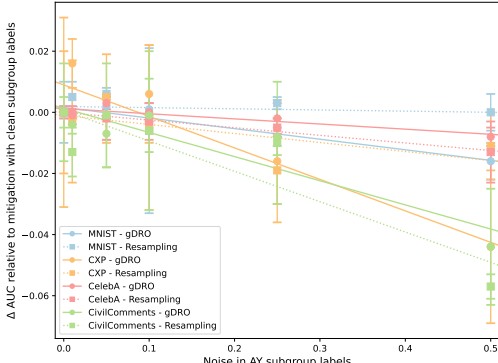

*Figure 4.* Noise in $(A, Y)$ subgroup labels leads to a degradation in AUC for gDRO and resampling. Each dot represents mean performance on the unbiased test set for a specific grouping, with error bars indicating the standard deviation across 3 random seeds.

using completely random subgroups generally worsens performance relative to ERM. This is most likely because the methods themselves are sub-optimal relative to ERM on the same distribution. This suggests that when one has no idea if the attributes for which they have annotations are related to possible biases, mitigation should not be conducted.

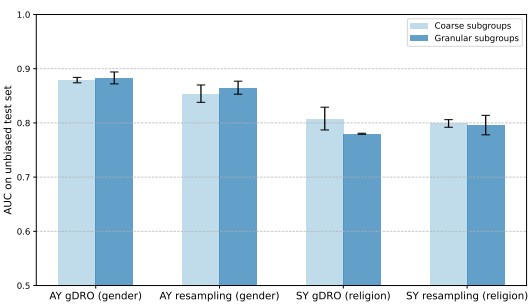

*Figure 3.* Performance on $\mathcal{P}_{unbiased}$ is similar for the coarse and granular subgroups in Civil_comments (mention of gender vs a specific male/female mention ($A$) and mention of any religion vs a specific mention of Christianity ($S$)). Error bars show standard deviation across 3 random seeds.

**Mitigation methods are relatively robust to noise.** As shown in Figure 4, performance of gDRO and resampling degrades when noise in the $(A, Y)$ annotations exceeds 10%. However, even under these conditions, these methods still outperform the baseline model, indicating that they are relatively robust to annotation noise affecting a minority of subgroup annotations. This aligns with findings from Awasthi et al. (2020) and (Stromberg et al., 2024) who explore the impact of noise in post-processing and last-layer retraining respectively. A similar trend is observed for DomainInd in the CXP dataset, although the effect is less clear for other model-based experiments, where performance frequently falls below the baseline across all noise levels (Figure A8).

**Random subgroups are generally detrimental.** Across the four datasets and mitigation methods we also find that

## 5.3. The ability to recover the unbiased distribution is key to mitigation success

We next aim to explain the observed variation in performance across different groupings. Inspired by (Zhou et al., 2021), we explore whether it is possible to recover the unbiased distribution by weighting the chosen subgroups. Our hypothesis is that the closer the weighted $\mathcal{P}_{train}$, which we denote $\mathcal{P}_{\text{train}}^{w}$, is to the test distribution $\mathcal{P}_{unbiased}$, the better the model will perform on $\mathcal{P}_{unbiased}$. This reflects the broader consensus in the generalisation literature that aligning train and test distributions (e.g., through methods like data balancing) can reduce generalisation error in non-i.i.d. settings (Mansour et al., 2009; Dong et al., 2024; Wang et al., 2023). For example, Ben-David et al. (Ben-David et al., 2010) show that for any hypothesis $h$, assuming the same labelling function in both source and target domains, the test error is bounded as follows:

$$\text{err}_{\text{test}}(h) \leq \text{err}_{\text{train}}(h) + D_{\text{TV}}(\mathcal{P}_{\text{train}}, \mathcal{P}_{\text{test}})$$

where $D_{\text{TV}}(\cdot, \cdot)$ denotes the total variation divergence.

We therefore choose to measure the divergence between $\mathcal{P}_{\text{train}}^{w}$ and $\mathcal{P}_{unbiased}$ for each subgrouping to see whether this is a predictor of performance on the unbiased distribution. We model the divergence as a Kullback-Leibler divergence following prior work which give upper bounds (Aminian et al., 2024; Masiha et al., 2021; Wu et al., 2024; Nguyen et al., 2022) and lower bounds (Masiha et al., 2021) for expected generalisation error. In our case, Pinsker's

inequality implies that test error can be bounded as:

$$\text{err}_{\text{unbiased}}(h) \leq \text{err}_{\text{train}}(h) + \sqrt{\frac{1}{2}\text{KL}(\mathcal{P}_{\text{train}}^w \parallel \mathcal{P}_{\text{unbiased}})}.$$

Since all our models reach a similarly low train error, we posit that the differences in upper bound are largely driven by the divergence between both distributions[1]. We explore whether the divergences achieved for each subgrouping correlate with generalisation error.

We assume that the difference between both distributions is attributable to differences in probabilities of sampling each $(Y, S, A)$ subgroup. Thus we represent each $\mathcal{P}_{\text{train}}^w$ as probability vectors in $\mathbb{R}^8$, as defined in Section 3.2. This yields an initial KL divergence $\text{KL}(\mathcal{P}_{\text{train}} \parallel \mathcal{P}_{\text{unbiased}}) \approx 0.527$. We next explore whether, for each possible subgrouping, by either resampling or gDRO, this divergence can be reduced. For resampling, each subgroup in the chosen subgrouping is resampled such that they are uniformly distributed. For instance, resampling across $(Y, S, A)$ would give $\mathcal{P}_{\text{train}}^w = \left[\frac{1}{8}, \frac{1}{8}, \dots, \frac{1}{8}\right]$. For gDRO, the distribution of the subgroups is learned during training by determining what weights to apply to each subgroup. We determine the theoretically optimal weights $w$ by solving a convex optimisation problem:

$$\min_{w \in \Delta^m} \text{KL}(\mathcal{P}_{\text{train}}^w \parallel \mathcal{P}_{\text{unbiased}}),$$

where $\Delta^k$ denotes the probability simplex over the selected $k$ subgroups. We present the divergences obtained for all groupings in Table 3 with full explanations and calculations provided in the Appendix E.

*Table 3.* Minimum $\text{KL}(\mathcal{P}_{\text{train}}^w \parallel \mathcal{P}_{\text{unbiased}})$ achievable by reweighting subgroups with gDRO and resampling.

| Grouping | KL divergence to $P_{unbiased}$ | |
| --- | --- | --- |
| | gDRO | Resampling |
| A | 0.527 | 0.527 |
| Y | 0.527 | 0.527 |
| S | 0.527 | 0.527 |
| AY | 0.113 | 0.113 |
| SY | 0.527 | 0.527 |
| YSA | 0.000 | 0.000 |
| SC/no-SC | 0.113 | 0.113 |
| $AY_8$ | 0.113 | 0.113 |
| $SY_8$ | 0.527 | 0.527 |
| Random | 0.527 | 0.527 |
| Noisy_$AY_{0.01}$ | 0.113 | 0.113 |
| Noisy_$AY_{0.05}$ | 0.113 | 0.114 |
| Noisy_$AY_{0.10}$ | 0.113 | 0.116 |
| Noisy_$AY_{0.25}$ | 0.114 | 0.131 |
| Noisy_$AY_{0.50}$ | 0.118 | 0.189 |

We observe a high correlation (between 0.73 and 0.97) between the minimum achievable divergence and the performance of each model across the four datasets for resampling

---

[1]We would ideally like to directly estimate generalisation under distribution shift (instead of just having an upper bound), but this would require very strong assumptions (Chuang et al., 2020).

and gDRO respectively (Figures 5, F10). This aligns with our hypothesis that the extent to which the unbiased distribution can be restored during training significantly influences generalisation performance. Of course, other factors may still impact performance, such as inherent differences in task difficulty across subgroups (Petersen et al., 2023). We also note that we are only able to do this analysis by assuming that any divergence between distributions is fully attributable to differences in $P(Y, S, A)$ and because we have full knowledge of how these distributions change at test time. These assumptions would rarely be validated in practical settings. Despite this, our results suggest that assessing whether an unbiased distribution can be recovered provides a useful starting point for defining subgroups in bias mitigation. Thus, **defining subgroups based on the cause of generalisation error may be more effective than simply defining subgroups based on observed disparities**.

These divergences also explain previous observations, such as the similarity of results when subgroup granularity increases (KL divergence is also unchanged), robustness to noise (KL divergences show little change), and the similarity between results for gDRO and resampling (also similar divergences). Moreover, it is interesting to note that incorporating $S$ into the $(A, Y)$ groups (to get $(Y, S, A)$) is the most optimal grouping, as we observe empirically for MNIST (Figure 2). Although $S$ is not involved in the $(A, Y)$ spurious correlation and not a cause for poor generalisation performance, simply reweighting the four $(A, Y)$ groups induces an unwanted correlation between $S$ and $Y$, since the $(A, Y)$ groups are imbalanced with respect to $S$ (Table 1).

### 5.4. Subgroup choice also impacts disparities

In addition to overall generalisation performance, we explore the impact of subgroup choice on disparities, specifically between $S$ subgroups (Table 4 and D9). Fairness results largely align with overall generalisation performance. We make the surprising observation that the best results in terms of disparities for a given grouping are not necessarily achieved by using that grouping in the bias mitigation process. For example, $(A, Y)$ subgroups show far better worst-group-accuracy and smaller accuracy gaps across $S$ subgroups than when the $S$ attribute is used for grouping (e.g., $S$, $SY$, $SY_8$) for both datasets and methods. The minimum accuracy is up to 0.21 higher and an accuracy gap up to 10 times smaller. This is because the $(A, Y)$ spurious correlation is causing the disparity in performance across $S$, and it can only be unlearned with certain groupings. This finding contradicts the general assumption that in order to improve fairness with respect to a certain subgroup, this subgroup should be used in the bias mitigation process (Mehrabi et al., 2021). To clarify, we do not advocate using different subgroups for evaluation, but rather that alternative subgroup definitions in *mitigation* may better improve fairness.

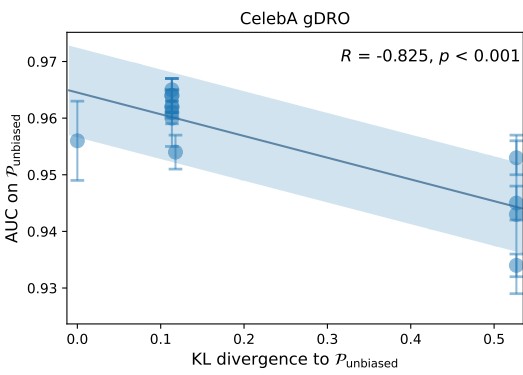
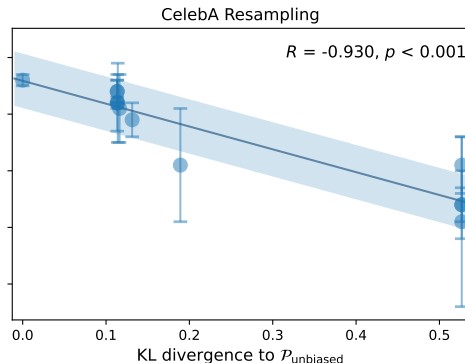

*Figure 5.* The ability to recover the unbiased test distribution ($\mathcal{P}_{unbiased}$) is a significant predictor of overall generalisation performance on $\mathcal{P}_{unbiased}$ for gDRO and resampling. Each dot represents mean AUC on the unbiased test set for a specific grouping, with error bars indicating the standard deviation across 3 random seeds. Pearson's correlation coefficients and associated p-values are also shown.

*Table 4.* Worst group accuracy and accuracy gap across S groups, averaged for gDRO and resampling. While some groupings lead to improved fairness relative to the baseline (green), others are detrimental (red). Most groupings involving $S$ have the opposite of their intended effect and decrease fairness with respect to $S$. We report mean and standard deviation across three random seeds.

| Grouping | min Acc. | Acc. gap |
|---|---|---|
| baseline | $0.705 \pm 0.025$ | $0.064 \pm 0.045$ |
| A | $0.696 \pm 0.019$ | $0.074 \pm 0.034$ |
| Y | $0.696 \pm 0.022$ | $0.077 \pm 0.037$ |
| S | $0.694 \pm 0.024$ | $0.072 \pm 0.039$ |
| AY | $0.779 \pm 0.02$ | $0.032 \pm 0.03$ |
| SY | $0.699 \pm 0.016$ | $0.074 \pm 0.026$ |
| YSA | $0.792 \pm 0.013$ | $0.02 \pm 0.022$ |
| SC/no-SC | $0.772 \pm 0.02$ | $0.04 \pm 0.03$ |
| $AY_8$ | $0.778 \pm 0.022$ | $0.041 \pm 0.032$ |
| $SY_8$ | $0.7 \pm 0.021$ | $0.072 \pm 0.032$ |
| Random | $0.693 \pm 0.02$ | $0.075 \pm 0.035$ |
| $Noisy\_AY_{0.50}$ | $0.741 \pm 0.016$ | $0.056 \pm 0.029$ |

## 5.5. Subgroup choice is similarly impactful in other settings

Finally, we conduct various additional experiments to verify that our results hold in less restrictive settings. We implement Just-train-twice (JTT), a method which does not require subgroup labels at training, but still requires some for model selection, and find that, again, its success is dependent on subgroup choice (Appendix C: Table C8 and Figure C9). We also repeat the MNIST experiments in a setting where the spurious correlation is weaker such that overall there are 77.5% spuriously correlated samples (instead of 87.5%). We find that while all results are higher overall, the same trends still appear, as shown in Figure G11. We also explore whether the relatively small size of the datasets we use (as we are constrained by the availability of each $(Y, S, A)$ combination) might impact our findings

by comparing our results on the full 60K MNIST dataset to a downsampled 7K version. We find largely similar trends, with an example shown in Figure G12.

## 5.6. Limitations and future work

This work is limited to a specific setting of bias i.e. spurious correlations, but many other types exist, including bias caused by differences in the manifestation of $Y$ across subgroups, differences in annotation of $Y$ across subgroups, and under-representation of certain subgroups. We hope that our research provides a solid foundation for further exploration into how subgroups should be defined for bias mitigation across a wide range of real bias settings. Another limitation is that, in practice, outside of (semi-)synthetic scenarios like ours, one often lacks extra annotations on other attributes, and is not perfectly aware of what might be causing underperformance. While we advocate for data collectors to gather as much metadata as possible to enable precise analyses, we recognise that this is not always feasible. Therefore, we encourage practitioners to carefully analyse their models' errors and thoughtfully investigate potential causes of underperformance before implementing bias mitigation strategies for specific subgroups.

## 6. Conclusions

To our knowledge, this is the first work to specifically and comprehensively consider how subgroup definition can impact existing bias mitigation methods. We demonstrate the extent to which certain subgroup definitions can "make or break" bias mitigation methods, and provide an explanation as to why this occurs. We urge practitioners to carefully consider possible causes of bias rather than indiscriminately applying bias mitigation techniques to any underperforming group. Our work enables more consistent and effective bias mitigation in real-world applications.

## Acknowledgements

A.A. is supported by the EPSRC grant number EP/S024093/1, the Centre for Doctoral Training SABS: R3, University of Oxford, and by GE Healthcare. C.J. is supported by Microsoft Research, EPSRC, and The Alan Turing Institute through a Microsoft PhD scholarship and a Turing PhD enrichment award. B.G. received support from the Royal Academy of Engineering as part of his Kheiron/RAEng Research Chair. The computational aspects of this research were supported by the Wellcome Trust Core Award Grant Number 203141/Z/16/Z and the NIHR Oxford BRC. The views expressed are those of the author(s) and not necessarily those of the NHS, the NIHR or the Department of Health. The authors would also like to thank the reviewers of this paper whose comments contributed to substantially improving the work.

## Impact Statement

This paper presents work whose goal is to advance the field of Machine Learning. There are many potential societal consequences of our work, none which we feel must be specifically highlighted here.

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

This appendix provides additional details and experiments that support the main text. It is structured as follows:

- A Supplementary experimental details on the datasets used, subgroups constructed, and model implementations.

- B Mitigation results for more bias mitigation methods: DomainInd and CFair.

- C Mitigation without subgroup labels (Just Train Twice).

- D Supplementary results for gDRO and resampling.

- E Explaining results through the divergence between $\mathcal{P}_{\text{train}}^w$ and $\mathcal{P}_{unbiased}$.

- F Correlation between KL divergence to the unbiased distribution and unbiased generalisation across all four datasets.

- G Ablations on strength of SC and size of dataset.

## A. Supplementary experimental details

### A.1. Dataset details

*Table 5.* Details on the datasets used for mitigation experiments.

| Dataset | MNIST | CheXPert | CelebA | Civil_comments |
|---|---|---|---|---|
| Y | Even/odd digit | Pleural effusion | Blonde hair | Toxicity |
| A | Background colour | Presence of a pacemaker | Perceived gender | Gender |
| S | Foreground colour | Sex | Smiling | Religion |
| Dataset size | 60000 | 3225 | 12500 | 8900 |

We downsample some of the datasets from their original size because we are constrained by the availability of each $(Y, S, A)$ combination. For example, for CheXPert, pacemaker annotations are only available for 4862 images, and we have to further downsample the dataset to make it balanced with respect to disease $(Y)$ and sex $(S)$.

### A.2. Subgroup construction

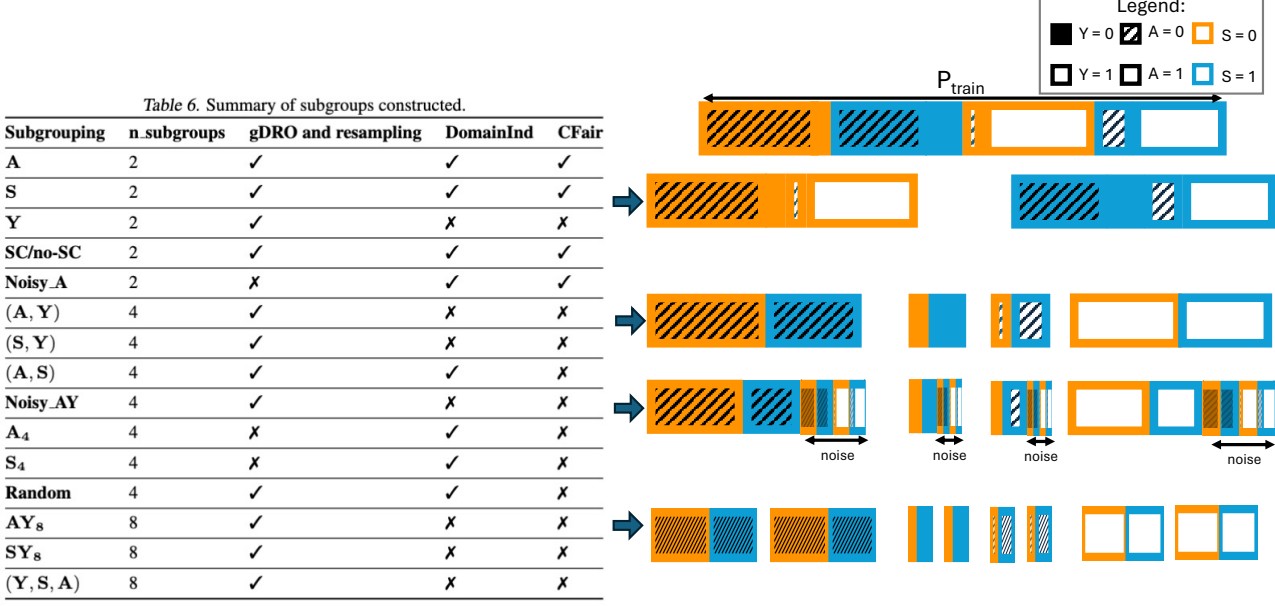

*Table 6.* Summary of subgroups constructed.

| Subgrouping | n_subgroups | gDRO and resampling | DomainInd | CFair |
|---|---|---|---|---|
| A | 2 | ✓ | ✓ | ✓ |
| S | 2 | ✓ | ✓ | ✓ |
| Y | 2 | ✓ | ✗ | ✗ |
| SC/no-SC | 2 | ✓ | ✓ | ✓ |
| Noisy_A | 2 | ✗ | ✓ | ✓ |
| (A, Y) | 4 | ✓ | ✗ | ✗ |
| (S, Y) | 4 | ✓ | ✗ | ✗ |
| (A, S) | 4 | ✓ | ✓ | ✗ |
| Noisy_AY | 4 | ✓ | ✗ | ✗ |
| A₄ | 4 | ✗ | ✓ | ✗ |
| S₄ | 4 | ✗ | ✓ | ✗ |
| Random | 4 | ✓ | ✓ | ✗ |
| AY₈ | 8 | ✓ | ✗ | ✗ |
| SY₈ | 8 | ✓ | ✗ | ✗ |
| (Y, S, A) | 8 | ✓ | ✗ | ✗ |

*Figure 6.* Subgroup construction for our experiments.

We show the subgroups used for each method and a visualisation of some example subgroups in Figure 6. For model-based methods (DomainInd and CFair), we do not use $Y$ to construct subgroups because for these methods to work best, each subgroup should contain both positive and negative classes. This is because methods like DomainInd and CFair learn representations for each subgroup separately. DomainInd trains a separate classifier for each subgroup, so it would not make sense to train a separate classification head for positive and negative classes. Similarly, CFair seeks to align subgroup representations, so it would not make sense to align representations of one subgroup containing only positive images to another subgroup containing only negative images, as this would defeat the point of training a discriminative classifier. On the other hand, for reweighting based methods, including the $Y$ in the subgroups helps to balance the final reweighted dataset with respect to class, and therefore improves results, especially in our case where the spurious correlation involves the class $Y$. This explains why we find that the subgroups which work well for DomainInd and CFair (e.g., $A$) are just a merged version of the ones which work well for gDRO and resampling (e.g., $(A, Y)$). To the best of our knowledge, no papers have explicitly discussed this distinction despite its practical importance.

### A.3. Implementation details

*Table 7.* Implementation details for all models.

| Training strategy | MNIST | CXP | CelebA | Civil_comments |
|---|---|---|---|---|
| Backbone | 2-layer CNN | DenseNet121 (Huang et al., 2017) | ResNet50 (He et al., 2016) | BERTClassifier (uncased) (Devlin et al., 2018) |
| Pre-training | None | ImageNet (Deng et al., 2009) | ImageNet (Deng et al., 2009) | Bookcorpus, Wikipedia (English) |
| Batch size | 128 | 256 | 256 | 32 |
| Image size | 3x28x28 | 3x299x299 | 3x256x256 | NA |
| Augmentation | Flip, rotation, Gaussian blur | Flip, rotation, color jitter, affine transformation, crop | Flip, rotation, color jitter, affine transformation, crop | None |
| Optimiser | Adam | Adam | Adam | AdamW |
| Loss | Binary cross-entropy | Binary cross-entropy | Binary cross-entropy | Binary cross-entropy |
| Learning rate | 0.001 | 0.0005 | 0.001 | 0.00005 |
| Learning scheduler | StepLR ($\gamma = 0.1$ and $\mu = 10$) | StepLR ($\gamma = 0.1$ and $\mu = 10$) | StepLR ($\gamma = 0.1$ and $\mu = 10$) | StepLR ($\gamma = 0.1$ and $\mu = 10$) |
| Weight decay | 0.0001 | 0.0001 | 0.0001 | 0.0001 |
| Max epochs | 50 | 100 (early stopping after 10) | 10 (early stopping after 5) | 10 (early stopping after 5) |

We conducted hyperparameter tuning on the baseline ERM model within the ranges below, and selected the model with the highest validation AUC. The choice of backbones was based on their strong performance in previous similar work (Irvin et al., 2019; Jain et al., 2024; Izmailov et al., 2022; Kirichenko et al., 2023; Idrissi et al., 2022).

- **Backbones for vision models**: ResNet18, ResNet50, DenseNet121 (not for MNIST images)

- **Batch size**: 32, 64, 128, 256, 512

- **Learning rate**: [1e-5:1e-3]

- **Weight decay**: [1e-5:1e-4]

We also specify additional hyperparameters for the mitigation methods: a step size of 0.01 and a size adjustment factor of 1 was used for gDRO, and a $\mu$ coefficient of 0.1 was used for the adversarial loss of CFair, following the MEDFAIR implementation (Zong et al., 2023).

# B. Results for more mitigation methods: DomainInd and CFair

Overall, CFair and DomainInd show less improvement on the unbiased test set than reweighting based-methods (Figure 7). Despite this, we still observe similar trends as for the reweighting based methods, such as $S$-related groups being clearly detrimental to performance. Learning independent models for $A$ groups also boosts performance for DomainInd in CXP, CelebA, and Civil_Comments, and while it does not significantly change performance relative to the baseline in MNIST, it is still higher than for any other grouping. Moreover, as shown in Figure 8, DomainInd appears sensitive to even low levels of noise, while CFair's performance is less degraded by noise in the $A$ subgroup labels (except for in MNIST, where both methods are ineffective, as performance stays close to the baseline for all subgroupings).

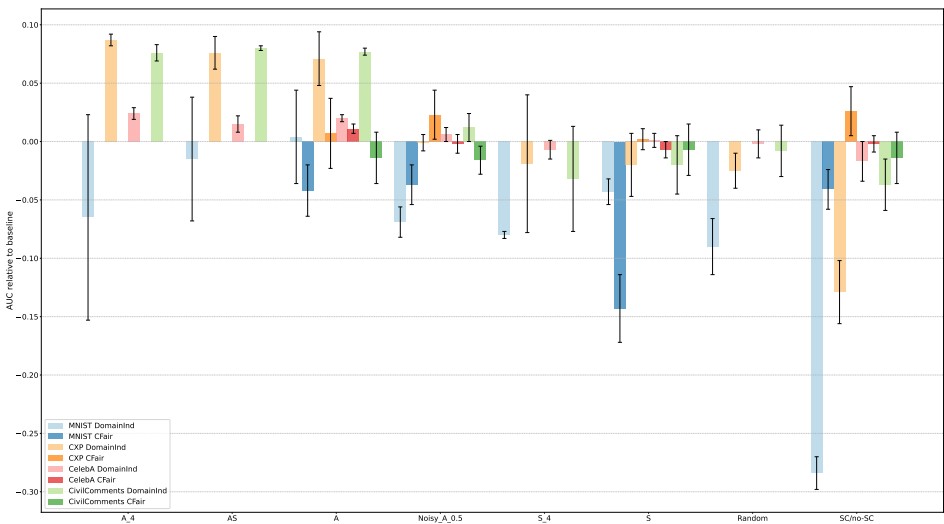

*Figure 7.* Relative performance on $\mathcal{P}_{unbiased}$ for different groupings in DomainInd and CFair across all four datasets. Similar trends to gDRO and resampling can be seen, where subgroupings constructed around $A$ generally improve performance as they prevent the SC from being learnt, while other subgroups are generally detrimental. Error bars indicate the standard deviation across 3 random seeds.

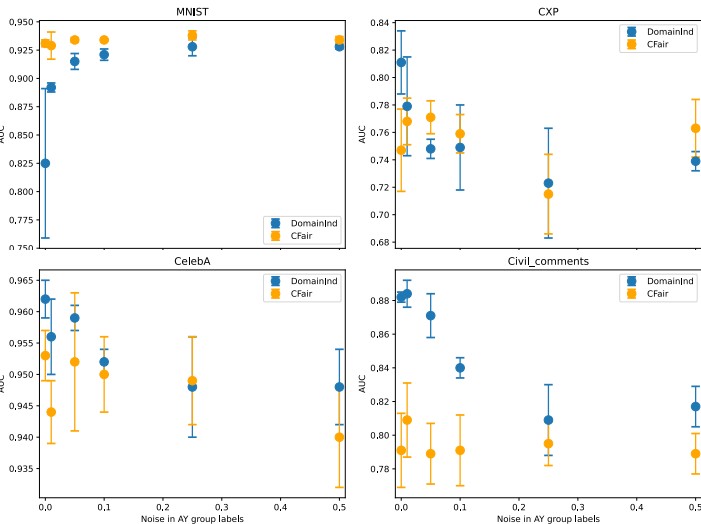

*Figure 8.* Effect of noise in $A$ subgroup labels on AUC for DomainInd and CFair. Each dot represents mean performance on the unbiased test set for a specific grouping, with error bars indicating the standard deviation across 3 random seeds.

## C. Subgroup discovery methods

We implement Just Train Twice (JTT) as proposed by Liu et al. (2021). JTT consists of a two-stage process, where first a standard ERM model is trained for several epochs, and then a second model that upweights the training examples that the first model misclassified is trained. Although it does not require subgroup labels for training, to select a final model to use (e.g., based on the JTT-specific hyperparameters), subgroup labels do need to be used.

As shown in Table 8 and Figure 9, we find that with validation subgroup labels to guide model and hyperparameter selection JTT performs mostly on par with our other methods, however, performance is again highly dependent on the choice of subgroups. When no subgroup annotations are used (i.e. model selection is done by overall validation accuracy), the method does not improve over ERM (except for on MNIST where JTT works remarkably effectively, most likely due to the simplicity of the task).

*Table 8.* Just train twice generalisation performance on unbiased test set across the four datasets is highly variable depending on the validation set subgroups used for model/hyper-parameter selection. We colour the experiments which improve over the baseline (no mitigation) in green and those do not in red. We report mean AUC and standard deviation across three random seeds.

| Subgroup | MNIST | CXP | CelebA | civil_comments |
|---|---|---|---|---|
| Baseline | $0.792 \pm 0.057$ | $0.740 \pm 0.002$ | $0.943 \pm 0.002$ | $0.805 \pm 0.017$ |
| SY | $0.89 \pm 0.002$ | $0.791 \pm 0.009$ | $0.947 \pm 0.005$ | $0.786 \pm 0.028$ |
| AY | $0.925 \pm 0.019$ | $0.791 \pm 0.009$ | $0.943 \pm 0.012$ | $0.831 \pm 0.004$ |
| A | $0.89 \pm 0.002$ | $0.695 \pm 0.018$ | $0.948 \pm 0.003$ | $0.786 \pm 0.011$ |
| AY_8 | $0.919 \pm 0.012$ | $0.791 \pm 0.009$ | $0.943 \pm 0.012$ | $0.812 \pm 0.021$ |
| S | $0.89 \pm 0.002$ | $0.791 \pm 0.009$ | $0.948 \pm 0.003$ | $0.786 \pm 0.028$ |
| SY_8 | $0.89 \pm 0.002$ | $0.734 \pm 0.035$ | $0.943 \pm 0.012$ | $0.786 \pm 0.028$ |
| Y | $0.89 \pm 0.002$ | $0.734 \pm 0.035$ | $0.947 \pm 0.005$ | $0.776 \pm 0.019$ |
| Noisy_AY_0.5 | $0.919 \pm 0.012$ | $0.705 \pm 0.042$ | $0.944 \pm 0.004$ | $0.831 \pm 0.004$ |
| Random | $0.925 \pm 0.019$ | $0.734 \pm 0.035$ | $0.948 \pm 0.003$ | $0.786 \pm 0.028$ |
| SC/no-SC | $0.936 \pm 0.009$ | $0.791 \pm 0.009$ | $0.943 \pm 0.012$ | $0.831 \pm 0.004$ |
| YSA | $0.922 \pm 0.006$ | $0.682 \pm 0.034$ | $0.943 \pm 0.012$ | $0.831 \pm 0.004$ |
| No val subgroups | $0.925 \pm 0.019$ | $0.734 \pm 0.035$ | $0.948 \pm 0.003$ | $0.786 \pm 0.028$ |

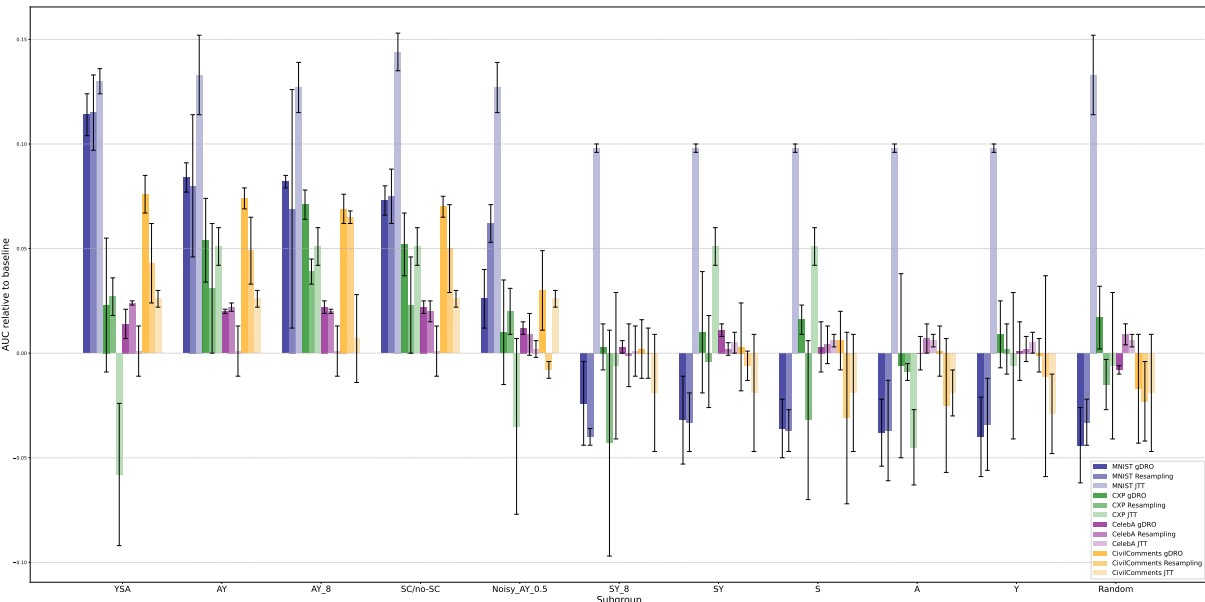

*Figure 9.* Performance on the unbiased test set in gDRO and resampling and JTT is highly dependent on the subgroups used. Bars represent overall change in AUC relative to the ERM baseline, with error bars indicating the standard deviation across 3 random seeds.

# D. Supplementary results for resampling and gDRO

Table 9. Worst group accuracy and accuracy gap across $S$ groups for gDRO and resampling. Some groupings improve fairness relative to the baseline, while others decrease worst-group-accuracy and increase the gap between the best- and worst-group. Mean and standard deviation across three random seeds are shown.

| Subgroup | MNIST gDRO min Acc. | MNIST gDRO Acc. gap | MNIST Resampling min Acc. | MNIST Resampling Acc. gap | CXP gDRO min Acc. | CXP gDRO Acc. gap | CXP Resampling min Acc. | CXP Resampling Acc. gap | CelebA gDRO min Acc. | CelebA gDRO Acc. gap | CelebA Resampling min Acc. | CelebA Resampling Acc. gap | Civil_comments gDRO min Acc. | Civil_comments gDRO Acc. gap | Civil_comments Resampling min Acc. | Civil_comments Resampling Acc. gap |
|---|---|---|---|---|---|---|---|---|---|---|---|---|---|---|---|---|
| Baseline | 0.599 ± 0.061 | 0.194 ± 0.112 | 0.599 ± 0.061 | 0.194 ± 0.079 | 0.633 ± 0.02 | 0.051 ± 0.026 | 0.633 ± 0.02 | 0.051 ± 0.026 | 0.861 ± 0.002 | 0.009 ± 0.018 | 0.861 ± 0.002 | 0.009 ± 0.016 | 0.726 ± 0.018 | 0.0 ± 0.048 | 0.726 ± 0.018 | 0.0 ± 0.035 |
| S | 0.574 ± 0.023 | 0.226 ± 0.04 | 0.583 ± 0.029 | 0.188 ± 0.032 | 0.645 ± 0.029 | 0.06 ± 0.03 | 0.624 ± 0.009 | 0.061 ± 0.02 | 0.875 ± 0.011 | 0.011 ± 0.024 | 0.857 ± 0.002 | 0.028 ± 0.009 | 0.716 ± 0.015 | 0.013 ± 0.034 | 0.72 ± 0.008 | 0.002 ± 0.018 |
| A | 0.782 ± 0.018 | 0.022 ± 0.038 | 0.744 ± 0.057 | 0.085 ± 0.058 | 0.698 ± 0.017 | 0.051 ± 0.032 | 0.655 ± 0.028 | 0.065 ± 0.042 | 0.898 ± 0.003 | 0.008 ± 0.005 | 0.893 ± 0.013 | 0.019 ± 0.015 | 0.79 ± 0.009 | 0.011 ± 0.016 | 0.773 ± 0.018 | 0.001 ± 0.031 |
| Y | 0.58 ± 0.015 | 0.192 ± 0.03 | 0.578 ± 0.023 | 0.193 ± 0.032 | 0.632 ± 0.027 | 0.077 ± 0.064 | 0.629 ± 0.007 | 0.079 ± 0.024 | 0.861 ± 0.016 | 0.018 ± 0.022 | 0.871 ± 0.022 | 0.018 ± 0.024 | 0.72 ± 0.021 | 0.013 ± 0.04 | 0.695 ± 0.021 | 0.0 ± 0.036 |
| SC/no-SC | 0.78 ± 0.009 | 0.014 ± 0.027 | 0.731 ± 0.097 | 0.095 ± 0.098 | 0.675 ± 0.006 | 0.101 ± 0.027 | 0.675 ± 0.025 | 0.068 ± 0.028 | 0.9 ± 0.011 | 0.01 ± 0.016 | 0.89 ± 0.008 | 0.019 ± 0.009 | 0.794 ± 0.011 | 0.004 ± 0.03 | 0.78 ± 0.008 | 0.02 ± 0.019 |
| SY | 0.575 ± 0.017 | 0.211 ± 0.028 | 0.579 ± 0.017 | 0.187 ± 0.026 | 0.652 ± 0.005 | 0.052 ± 0.039 | 0.613 ± 0.032 | 0.06 ± 0.052 | 0.863 ± 0.022 | 0.024 ± 0.033 | 0.864 ± 0.023 | 0.023 ± 0.027 | 0.721 ± 0.03 | 0.007 ± 0.043 | 0.685 ± 0.049 | 0.012 ± 0.068 |
| AY | 0.588 ± 0.025 | 0.209 ± 0.031 | 0.575 ± 0.01 | 0.185 ± 0.013 | 0.648 ± 0.011 | 0.082 ± 0.02 | 0.611 ± 0.059 | 0.045 ± 0.084 | 0.863 ± 0.004 | 0.009 ± 0.019 | 0.864 ± 0.02 | 0.013 ± 0.035 | 0.717 ± 0.022 | 0.017 ± 0.034 | 0.731 ± 0.015 | 0.011 ± 0.016 |
| Random | 0.577 ± 0.019 | 0.21 ± 0.027 | 0.578 ± 0.026 | 0.199 ± 0.03 | 0.644 ± 0.005 | 0.054 ± 0.005 | 0.632 ± 0.012 | 0.075 ± 0.029 | 0.858 ± 0.026 | 0.019 ± 0.054 | 0.861 ± 0.011 | 0.027 ± 0.022 | 0.721 ± 0.018 | 0.008 ± 0.028 | 0.702 ± 0.055 | 0.022 ± 0.075 |
| Noisy_$AY_{001}$ | 0.771 ± 0.005 | 0.035 ± 0.022 | 0.735 ± 0.056 | 0.093 ± 0.057 | 0.687 ± 0.015 | 0.076 ± 0.015 | 0.66 ± 0.026 | 0.071 ± 0.034 | 0.891 ± 0.006 | 0.022 ± 0.011 | 0.89 ± 0.005 | 0.024 ± 0.006 | 0.785 ± 0.004 | 0.013 ± 0.014 | 0.763 ± 0.016 | 0.001 ± 0.025 |
| Noisy_$AY_{005}$ | 0.768 ± 0.011 | 0.041 ± 0.028 | 0.759 ± 0.035 | 0.084 ± 0.037 | 0.692 ± 0.018 | 0.073 ± 0.026 | 0.664 ± 0.013 | 0.073 ± 0.044 | 0.899 ± 0.009 | 0.007 ± 0.014 | 0.892 ± 0.016 | 0.017 ± 0.016 | 0.775 ± 0.009 | 0.026 ± 0.021 | 0.774 ± 0.012 | 0.001 ± 0.023 |
| Noisy_$AY_{010}$ | 0.76 ± 0.01 | 0.048 ± 0.033 | 0.686 ± 0.023 | 0.148 ± 0.03 | 0.686 ± 0.022 | 0.079 ± 0.022 | 0.662 ± 0.008 | 0.06 ± 0.018 | 0.897 ± 0.006 | 0.013 ± 0.01 | 0.898 ± 0.01 | 0.014 ± 0.01 | 0.783 ± 0.017 | 0.026 ± 0.031 | 0.766 ± 0.029 | 0.006 ± 0.037 |
| Noisy_$AY_{025}$ | 0.754 ± 0.018 | 0.042 ± 0.035 | 0.752 ± 0.021 | 0.089 ± 0.022 | 0.667 ± 0.021 | 0.068 ± 0.039 | 0.64 ± 0.021 | 0.074 ± 0.021 | 0.896 ± 0.009 | 0.006 ± 0.021 | 0.886 ± 0.008 | 0.013 ± 0.011 | 0.788 ± 0.007 | 0.005 ± 0.009 | 0.767 ± 0.023 | 0.006 ± 0.03 |
| Noisy_$AY_{050}$ | 0.686 ± 0.029 | 0.111 ± 0.059 | 0.695 ± 0.013 | 0.143 ± 0.017 | 0.656 ± 0.011 | 0.077 ± 0.029 | 0.66 ± 0.018 | 0.057 ± 0.029 | 0.879 ± 0.01 | 0.021 ± 0.015 | 0.873 ± 0.009 | 0.019 ± 0.018 | 0.754 ± 0.029 | 0.007 ± 0.042 | 0.724 ± 0.014 | 0.011 ± 0.023 |
| $AY_8$ | 0.579 ± 0.02 | 0.193 ± 0.037 | 0.584 ± 0.025 | 0.185 ± 0.028 | 0.642 ± 0.004 | 0.093 ± 0.025 | 0.621 ± 0.019 | 0.062 ± 0.034 | 0.84 ± 0.007 | 0.018 ± 0.031 | 0.878 ± 0.011 | 0.015 ± 0.015 | 0.699 ± 0.05 | 0.015 ± 0.078 | 0.699 ± 0.028 | 0.02 ± 0.03 |
| $SY_8$ | 0.749 ± 0.024 | 0.046 ± 0.053 | 0.74 ± 0.034 | 0.082 ± 0.035 | 0.681 ± 0.024 | 0.08 ± 0.027 | 0.658 ± 0.018 | 0.068 ± 0.037 | 0.893 ± 0.005 | 0.017 ± 0.01 | 0.897 ± 0.018 | 0.014 ± 0.022 | 0.792 ± 0.008 | 0.009 ± 0.027 | 0.768 ± 0.025 | 0.005 ± 0.026 |
| YSA | 0.817 ± 0.019 | 0.014 ± 0.024 | 0.819 ± 0.021 | 0.03 ± 0.026 | 0.688 ± 0.016 | 0.033 ± 0.019 | 0.673 ± 0.009 | 0.05 ± 0.036 | 0.89 ± 0.009 | 0.005 ± 0.014 | 0.901 ± 0.007 | 0.008 ± 0.009 | 0.795 ± 0.012 | 0.004 ± 0.019 | 0.754 ± 0.011 | 0.021 ± 0.035 |

# E. KL divergence between $\mathcal{P}_{\text{train}}^w$ and $\mathcal{P}_{\text{unbiased}}$

## E.1. Method overview

Our objective is to measure the minimum KL divergence which can be achieved to $\mathcal{P}_{unbiased}$ by partitioning $\mathcal{P}_{\text{train}}$ into subgroups and re-weighting these subgroups.

We define each distribution as probability vectors in $\mathbb{R}^8$ with each element corresponding to the probability of sampling one $(Y, S, A)$ subgroup. The unbiased distribution is defined to be uniform, i.e., $\mathcal{P}_{\text{unbiased}} = \left[\frac{1}{8}, \frac{1}{8}, \ldots, \frac{1}{8}\right]$, while $\mathcal{P}_{\text{train}} = \left[\frac{0.95}{4}, \frac{0.05}{4}, \frac{0.8}{4}, \frac{0.2}{4}, \frac{0.05}{4}, \frac{0.95}{4}, \frac{0.2}{4}, \frac{0.8}{4}\right]$.

Initially, $\text{KL}(\mathcal{P}_{\text{train}} \| \mathcal{P}_{\text{unbiased}}) \approx 0.527$. Our aim is to see whether different subgroupings can reduce this divergence.

Let $\mathcal{G} = \{G_1, \ldots, G_k\}$ be a partition of the 8 atomic subgroups into $k$ disjoint groups. For a set of weights $w = [w_1, \ldots, w_k] \in \Delta^k$ over these groups, we define a new weighted distribution $\mathcal{P}_{\text{train}}^w \in \mathbb{R}^8$ as follows:

$$\mathcal{P}_{\text{train}}^w[j] = w_i \cdot \frac{\mathcal{P}_{\text{train}}[j]}{\sum_{l \in G_i} \mathcal{P}_{\text{train}}[l]} \quad \text{for } j \in G_i.$$

Let the atomic subgroup indices correspond to $(Y, S, A)$ combinations in order $[0, 1, 2, 3, 4, 5, 6, 7]$. For the subgroups we constructed, we therefore have:

- $Y$: $\{\{0, 1, 2, 3\}, \{4, 5, 6, 7\}\}$

- $A$: $\{\{0, 2, 4, 6\}, \{1, 3, 5, 7\}\}$

- $S$: $\{\{0, 1, 4, 5\}, \{2, 3, 6, 7\}\}$

- $(A, Y)$: $\{\{0, 2\}, \{1, 3\}, \{4, 6\}, \{5, 7\}\}$

- $(S, Y)$: $\{\{0, 1\}, \{2, 3\}, \{4, 5\}, \{6, 7\}\}$

- $(Y, S, A)$: $\{0\}, \{1\}, \{2\}, \{3\}, \{4\}, \{5\}, \{6\}, \{7\}$

- (SC,no-SC): $\{\{0, 2, 5, 7\}, \{1, 3, 4, 6\}\}$

- Random: $\{\{0, 1, 2, 3, 4, 5, 6, 7\}\}$

- $AY_8$: $\{\{0, 2\}, \{0, 2\}, \{1, 3\}, \{1, 3\}, \{4, 6\}, \{4, 6\}, \{5, 7\}, \{5, 7\}\}$

- $SY_8$: $\{\{0, 1\}, \{0, 1\}, \{2, 3\}, \{2, 3\}, \{4, 5\}, \{4, 5\}, \{6, 7\}, \{6, 7\}\}$

For the Noisy$_{AY}$ subgroups, we have the same construction as the $(A, Y)$ subgroups, except that $b$ fraction of the $(A, Y)$ subgroup annotations are misannotated following the original $\mathcal{P}_{train}$ distribution, while the remaining $(1 - b)$ are consistent with the original $(A, Y)$ subgroups.

Figure 6 provides an illustration of some of these subgroups.

## E.2. Resampling

For resampling, these weights are uniformly distributed such that $w = [1/k, ..., 1/k] \in \Delta^k$. We can therefore calculate $\mathcal{P}_{\text{train}}^w$) for each grouping keeping the relative proportions of the $(Y, S, A)$ combinations constant within a group. For example, for $Y$, the two groups have probabilities which sum to $\mathcal{P}_{G_1} = \mathcal{P}_{G_2} = \frac{0.95}{4} + \frac{0.05}{4} + \frac{0.8}{4} + \frac{0.2}{4} = \frac{1}{2}$, so the relative proportions within the two subgroups are $[\frac{\frac{0.95}{4}}{\mathcal{P}_{G_1}}, \frac{\frac{0.05}{4}}{\mathcal{P}_{G_1}}, \frac{\frac{0.8}{4}}{\mathcal{P}_{G_1}}, \frac{\frac{0.2}{4}}{\mathcal{P}_{G_1}}, \frac{\frac{0.05}{4}}{\mathcal{P}_{G_2}}, \frac{\frac{0.95}{4}}{\mathcal{P}_{G_2}}, \frac{\frac{0.2}{4}}{\mathcal{P}_{G_2}}, \frac{\frac{0.8}{4}}{\mathcal{P}_{G_2}},]$. By multiplying these probabilities by $w = [\frac{1}{2}, \frac{1}{2}]$, we get $\mathcal{P}_{\text{train}}^w = \mathcal{P}_{\text{train}}$, so $\text{KL}(\mathcal{P}_{\text{train}}^w \| \mathcal{P}_{\text{unbiased}}) \approx 0.527$.

We proceed in this way for all subgroups, and obtain the divergences detailed in Table 3.

### E.3. gDRO

For gDRO, the groups are the same but the weights are learned during training. Therefore we determine the weights which could in theory be achieved to give the lowest KL divergence [2]. To do this, we reframe the problem as a convex optimisation problem where we minimize the following objective:

$$\min_{w \in \Delta^k} \quad \mathrm{KL}\big(\mathcal{P}^w_{\text{train}} \parallel \mathcal{P}_{\text{unbiased}}\big)$$

$$\text{subject to} \quad w_i > 0, \quad \sum_{i=1}^{k} w_i = 1.$$

For each subgrouping, we calculate the relative probabilities within a subgroup (as for resampling) and then use `scipy.optimize.minimize` and `scipy.special.rel_entr` to determine the optimal weight vector subject to the constraints above.

For example, for $(A, Y)$ subgroups, $\mathcal{P}_{G_1} = \mathcal{P}_{G_4} = \frac{0.95}{4} + \frac{0.8}{4} = \frac{1.75}{4}$ and $\mathcal{P}_{G_2} = \mathcal{P}_{G_3} = \frac{0.05}{4} + \frac{0.2}{4} = \frac{0.25}{4}$, so the relative proportions within the two subgroups are $[\frac{\frac{0.95}{4}}{\mathcal{P}_{G_1}}, \frac{\frac{0.05}{4}}{\mathcal{P}_{G_2}}, \frac{\frac{0.8}{4}}{\mathcal{P}_{G_1}}, \frac{\frac{0.2}{4}}{\mathcal{P}_{G_2}}, \frac{\frac{0.05}{4}}{\mathcal{P}_{G_3}}, \frac{\frac{0.95}{4}}{\mathcal{P}_{G_4}}, \frac{\frac{0.2}{4}}{\mathcal{P}_{G_3}}, \frac{\frac{0.8}{4}}{\mathcal{P}_{G_4}},]$. Minimisation gives $w = [\frac{1}{4}, \frac{1}{4}, \frac{1}{4}, \frac{1}{4}]$, yielding $\mathcal{P}^w_{\text{train}} = [0.136, 0.050, 0.114, 0.200, 0.050, 0.136, 0.200, 0.114]$, and $\mathrm{KL}(\mathcal{P}^w_{\text{train}} \parallel \mathcal{P}_{\text{unbiased}}) \approx 0.113$. Complete results for all groupings are presented in Table 3. We often find that these weights correspond to those used for resampling.

---

[2]We note that these weights may not necessarily be attained in practice by all gDRO models because they are not specifically trained with this objective (although in our setting, minimising KL divergence to the unbiased test set should be a reasonable proxy for minimising worst-group loss). Also, stochasticity in training, optimisation challenges, and inherent difficulties in the task across subgroups may also affect the attainment of this optimum. Despite this, we believe doing this calculation still provides an important indication of the *potential* effectiveness of a chosen subgrouping.

## F. Correlation between KL divergence and unbiased generalisation

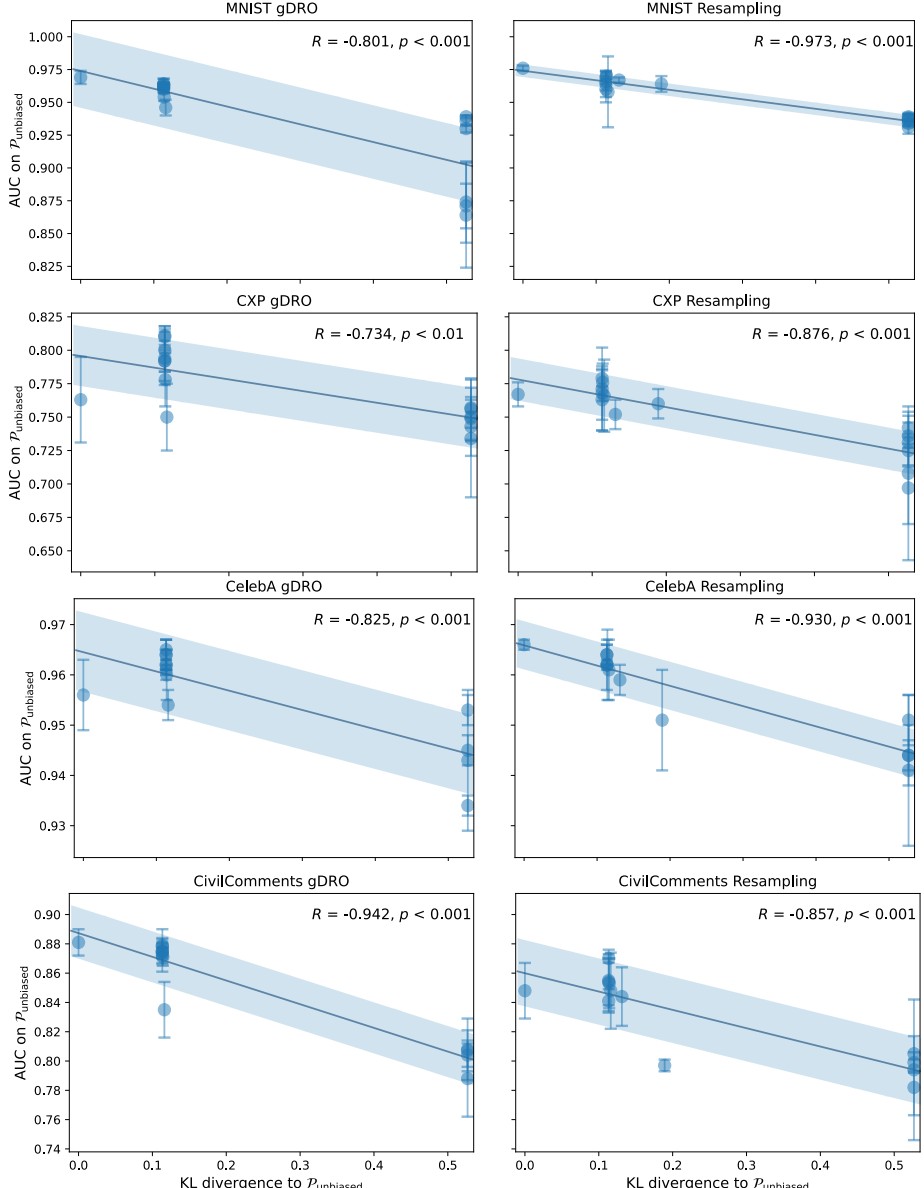

*Figure 10.* Test AUC is highly correlated with the minimum achievable KL divergence between $\mathcal{P}_{\text{train}}^{w}$ and $\mathcal{P}_{\text{unbiased}}$ across all four datasets for gDRO and resampling in CXP. Each dot represents mean performance on the unbiased test set for a specific grouping, with error bars indicating the standard deviation across 3 random seeds.

# G. Various ablations

### G.1. MNIST results with a weaker spurious correlation

To verify that our results still hold in settings where the spurious correlation is weaker, we re-generate the MNIST dataset in the exact same way, except that $P(Y = 0, A = 0 \mid S = 0) = P(Y = 1, A = 1 \mid S = 0) = 0.85$ and $P(Y = 0, A = 0 \mid S = 0) = P(Y = 1, A = 1 \mid S = 0) = 0.70$, such that overall there are 77.5% spuriously correlated samples, instead of 87.5%. We repeat the same experiments and find that, while all results are higher overall, the same trends still appear. Notably, we identify a significant correlation between the minimum achievable KL divergence to $\mathcal{P}_{unbiased}$ and the overall performance on $\mathcal{P}_{unbiased}$, as shown in Figure G11. This suggests that subgroup choice is an important factor in less extreme settings of bias as well.

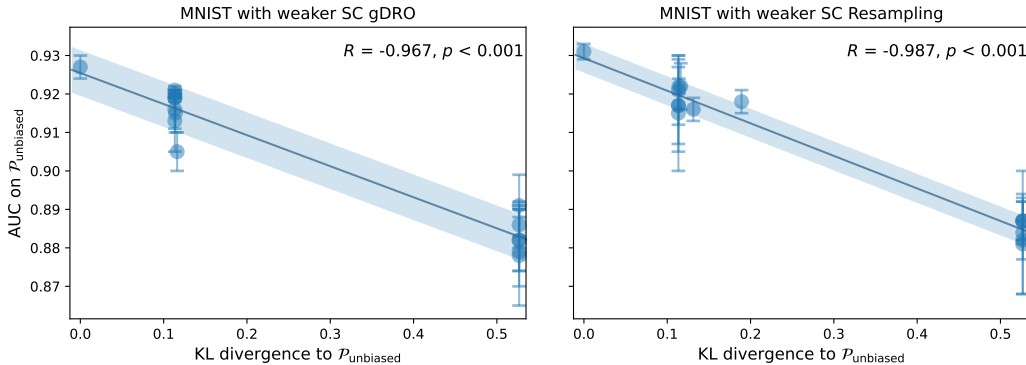

*Figure 11.* Relationship between AUC and the minimum achievable distance to the unbiased test distribution ($P_{unbiased}$) for gDRO and resampling in MNIST with a weaker spurious correlation. Each dot represents mean performance on the unbiased test set for a specific grouping, with error bars indicating the standard deviation across 3 random seeds.

### G.2. MNIST experiments with a smaller dataset

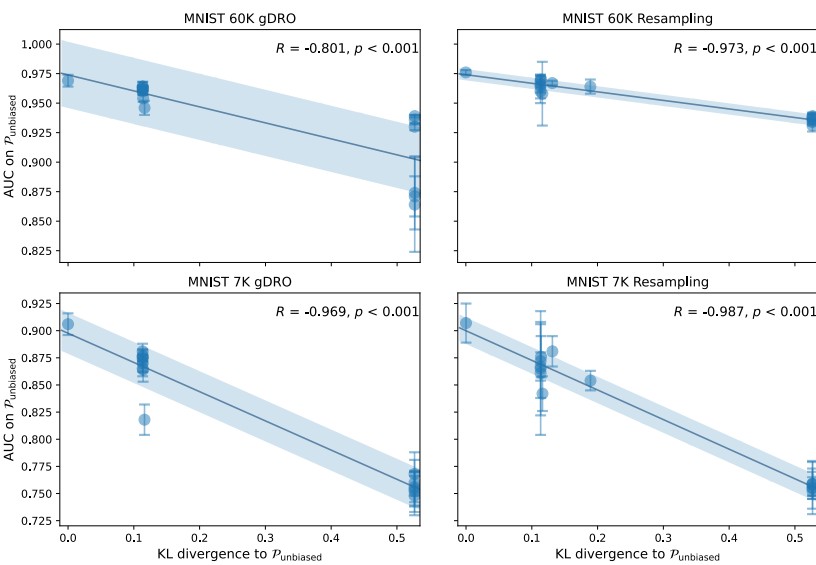

*Figure 12.* Relationship between AUC and the minimum achievable KL divergence to the unbiased test distribution ($P_{unbiased}$) for gDRO and resampling in MNIST with a downsampled dataset. Trends appear similar across both dataset sizes, suggesting that the results on the other three datasets would hold had we been able to use a larger subgroup-annotated dataset. Each dot represents mean performance on the unbiased test set for a specific grouping, with error bars indicating the standard deviation across 3 random seeds.

