# OpenReview forum: "Subgroups Matter for Robust Bias Mitigation"
_ICML.cc/2025/Conference — ICML 2025 poster_

### Official Review · Reviewer_6xGC · 2025-02-25

**Overall Recommendation:** 3

**Summary:**

This paper studies how the definition of subgroups affects the efficacy of bias mitigation techniques for spurious correlations. A causal graph approach is introduced to formalize correlations between classes, attributes, and subgroups, which are then manipulated to study AUC with respect to an ERM baseline on two semi-synthetic image datasets. The results show that subgroups have a large impact on bias mitigation, including leading to worse outcomes than ERM, and that increasing granularity has no significant effect on performance. Finally, the paper proposes the distance from an unbiased distribution achieved by resampling as an effective generalization measure.

## Update after rebuttal
Although I initially recommended rejection, the authors provided a very detailed and comprehensive rebuttal which addressed all of my critiques. Most importantly, the paper is substantially improved with the addition of more challenging datasets across a variety of domains, as well as a reduction in theoretical overclaims and situation of the theory within established generalization results. Therefore, I now recommend a weak accept.

**Claims And Evidence:**

Overall, the presented evidence supports the paper’s empirical claims on the two datasets studied. My main concerns are a limited scope of evaluation (discussed in Methods and Evaluation Criteria) and overclaimed theoretical results (discussed in Theoretical Claims).

**Essential References Not Discussed:**

1. Figure 3 of this paper studies performance of gDRO and resampling under noise in subgroup labels. A recent paper which should be discussed in this section is [3], which provides evidence of degradation of the WGA of resampling (called upweighting by [3]) under noise in the subgroup labels (called domain labels by [3]). Overall, the novelty of the experiments in Figure 3 is somewhat limited in comparison to [3].

2. It would also be beneficial to discuss references which study definitions of subgroups across spurious features. For example, [1] examine a dataset with multiple spurious correlations and show that defining subgroups with respect to bias type (e.g., age or gender) has a large impact on robustness method performance. Moreover, both [1] and [2] show that mitigating spurious correlation with respect to one subgroup definition may actually exacerbate bias with respect to a different definition.

[1] Kim et al. Improving Robustness to Multiple Spurious Correlations by Multi-Objective Optimization. ICML 2024.

[2] Li et al. A Whac-A-Mole Dilemma: Shortcuts Come in Multiples Where Mitigating One Amplifies Others. CVPR 2023.

[3] Stromberg et al. Robustness to Subpopulation Shift with Domain Label Noise via Regularized Annotation of Domains. TMLR 2024.

**Experimental Designs Or Analyses:**

The experimental design, particularly with respect to the subgroup generation step, is clearly explained and well-justified. The additional specifications and results in the Appendix are welcomed.

**Methods And Evaluation Criteria:**

1. My main concern regarding evaluation is the lack of challenging datasets across a variety of evaluation domains. This paper focuses on semi-synthetic versions of the MNIST and CheXPert datasets. The simplicity of MNIST lends itself to more of a sanity check than a benchmark. Moreover, while CheXPert is a good real-world dataset, the paper downsamples it to only 2330 training images (originally 200K+). Overall, the evaluation in this paper is below the standard in spurious correlations literature of at least 3-4 datasets across both vision and language tasks (e.g., [2, 3, 4, 7]), making it difficult to assess the generality of its conclusions.

2. This paper provides results on resampling, but does not mention the downsampling method (wherein data from larger groups is removed prior to training) proposed by [1] (cited by the authors). Downsampling is often superior to resampling, as resampling WGA may collapse over training [1, 5, 6]. In particular, it’s not clear whether the poor performance of resampling in Figure 2 is due to group definitions or the suboptimality of resampling as a technique.

[1] Idrissi et al. Simple data balancing achieves competitive worst-group-accuracy. CLeaR 2022.

[2] Izmailov et al. On Feature Learning in the Presence of Spurious Correlations. NeurIPS 2022.

[3] Kirichenko et al. Last Layer Re-Training is Sufficient for Robustness to Spurious Correlations. ICLR 2023.

[4] Koh et al. WILDS: A Benchmark of in-the-Wild Distribution Shifts. ICML 2021.

[5] LaBonte et al. The Group Robustness is in the Details: Revisiting Finetuning under Spurious Correlations. NeurIPS 2024.

[6] Stromberg et al. Robustness to Subpopulation Shift with Domain Label Noise via Regularized Annotation of Domains. TMLR 2024.

[7] Zhou et al. Examining and Combating Spurious Features under Distribution Shift. ICML 2021.

**Other Comments Or Suggestions:**

I have a minor issue with the paper’s assertion that “approaches [which do not utilize group annotations] often fail to consistently outperform traditional subgroup-based methods and are not
widely adopted” (line 121-122). The references cited in this paragraph are somewhat outdated, and the statement minimizes the progress made by more recent methods. For example, [1, 5, 9] are competitive with gDRO [8] and [4, 7] are competitive with DFR [2]. As far as adoption of such methods, at least [6] (cited in this paper) is considered widely adopted, though nowadays far from state-of-the-art.

Regarding the granularity discussion in Section 5.2, an interesting case study may be the CivilComments dataset [3]. There are two versions of this dataset in the literature: one with four groups, where the identity categories (male, female, LGBT, black, white, Christian, Muslim, or other religion) are collapsed into a single spurious feature, and one which uses the un-collapsed identity categories. It would be interesting to see if the granularity findings hold across these two versions of CivilComments, and whether results in the literature are consistent.

Below, I’ve included a list of typos or sentences where more clarification is needed. Also, the plots are pixelated and sometimes hard to read; it would be helpful to increase the DPI.

1. Line 88: Subgroup typo.

2. Line 135: $\mathbb{E}\_P$ should be $\mathbb{E}_{\mathcal{P}}$?

3. Equation 2: $\mathcal{G}$ is not defined; I assume it is the set of groups.

4. Equation 3: The average over $m$ is meaningless here as one is already taking the expectation over $\mathcal{P}_{train}$; the index $i$ is unused within the sum.

5. Equation 3: $P_{train_G}$ should be $P_{train_g}$?

6. Lines 205-206: =0 and =1 should be in math mode.

7. Section 5.3 and 5.5: \mid should be used for conditional probabilities.

[1] Han et al. Improving Group Robustness on Spurious Correlation Requires Preciser Group Inference. ICML 2024.

[2] Kirichenko et al. Last Layer Re-Training is Sufficient for Robustness to Spurious Correlations. ICLR 2023.

[3] Koh et al. WILDS: A Benchmark of in-the-Wild Distribution Shifts. ICML 2021.

[4] LaBonte et al. Towards Last-layer Retraining for Group Robustness with Fewer Annotations. NeurIPS 2023.

[5] Liu et al. Avoiding spurious correlations via logit correction. ICLR 2023.

[6] Liu et al. Just train twice: Improving group robustness without training group information. ICML 2021.

[7] Qiu et al. Simple and Fast Group Robustness by Automatic Feature Reweighting. ICML 2023.

[8] Sagawa et al. Distributionally Robust Neural Networks for Group Shifts: On the Importance of Regularization for Worst-Case Generalization. ICLR 2020.

[9] Zhang et al. Correct-N-Contrast: A Contrastive Approach for Improving Robustness to Spurious Correlations. ICML 2022.

**Other Strengths And Weaknesses:**

One strength of the paper is its comprehensiveness in investigating as many as 15 different types of subpopulation groupings. The granularity and random groupings are particularly interesting.

One weakness (as noted by the authors in the Limitations section) is that the paper focuses on the influence of subgroup definition within bias mitigation methods that utilize group annotations, which is a somewhat unrealistic setting. Nevertheless, since understanding methods that utilize group annotations is likely a prerequisite for understanding methods that infer subgroups, the study is still worthwhile.

**Questions For Authors:**

As mentioned in the “Theoretical Claims” section, why was mean absolute error chosen for the minimum distance metric instead of mean squared error, a worst-group metric, or KL divergence?

**Relation To Broader Scientific Literature:**

Multiple other works have proposed rethinking the definitions of spurious correlations (e.g., [5]) as well as the definitions of subgroups [2, 3]. However, to my knowledge this is the only paper to evaluate subgroup definitions in the context of the causal learning literature (e.g., [1, 4]).

[1] Jones et al. Rethinking fair representation learning for performance-sensitive tasks. ICLR 2025.

[2] Kim et al. Improving Robustness to Multiple Spurious Correlations by Multi-Objective Optimization. ICML 2024.

[3] Li et al. A Whac-A-Mole Dilemma: Shortcuts Come in Multiples Where Mitigating One Amplifies Others. CVPR 2023.

[4] Schrouff et al. Mind the graph when balancing data for fairness or robustness. NeurIPS 2024.

[5] Yang et al. Change is Hard: A Closer Look at Subpopulation Shift. ICML 2023.

**Theoretical Claims:**

1. The statement from the introduction that this paper “[provides] a theoretical explanation for the differences observed based on the minimum distance between the group-weighted biased distribution and the unbiased test distribution” is overclaimed. While Pearson coefficients are provided, no theoretical result is proven which connects the proposed distance metric to model performance (i.e., a generalization bound).

2. The proposed distance metric is insufficiently justified independently of [1]. It makes intuitive sense -- the lower the distance metric, the better the “optimal” resampling will be. However, the distance from the resampling distribution to $\mathcal{P}_{unbiased}$ is chosen to be the mean absolute error without justification. Why not mean squared error, the absolute or squared error on the worst group only, or a probability divergence such as KL divergence?

[1] Zhou et al. Examining and Combating Spurious Features under Distribution Shift. ICML 2021.

---

> ### Author Rebuttal · Authors · 2025-04-01
>
> We thank the reviewer for their thoughtful and comprehensive review. We agree with the comments and feel they have helped us substantially improve the paper. We have made clarifications to the manuscript and run >200 additional experiments. We include details of our changes below and share all results in this [folder](https://rb.gy/xebqmp). We hope this will give the reviewer confidence to raise their score.
>
> **1. Lack of challenging datasets across domains**
>
> We agree that expanding the scope of evaluation would strengthen the results. To address this, **we have expanded our evaluation to include two additional datasets**: civil_comments (text), and CelebA (natural images). Encouragingly, these datasets show consistent trends and help to bolster the generalisability of our conclusions. Please see details on the updated setup in Tab A3 and key results in Figs 2 and 5.
>
> **2. Downsampling** is often superior to resampling. Is resampling’s poor performance due to group definitions or its suboptimality?
>
> We should clarify we do not think the takeaway should be that resampling performs poorly, but that it performs *variably* depending on the subgroups used. In Fig 2, we observe that it performs quite well for some subgroups (e.g. it increases AUC by >0.10 relative to ERM on MNIST with $YAS$ groups and by >0.04 on CXP with $AY_8$ groups).
>
> However, we agree that given the literature it would be interesting to see whether similar results hold for downsampling. **We implement downsampling** as described in [1] and find that results are closely aligned for CXP, CelebA, and civil_comments (Fig C2). Downsampling actually underperforms on MNIST, so we maintain resampling for its consistency.
>
> **3. Granularity findings across two versions of civil_comments**
>
> We agree this would be a very interesting case study. As we were unable to find this in the literature, we did this experiment by splitting the attribute $A$ (any mention of gender) into two more subgroups (mentions male/does not) and likewise for $S$ (religion) into Christian/non-Christian. We found that performance was very similar across the granular and coarse groupings, in line with our results on synthetic granular groups (Fig 3). We are grateful for this suggestion and think this is a nice real-world and practical insight to strengthen our paper.
>
> **4. Downsampling CheXPert dataset**
>
> This is a fair point, however, we note that pacemaker annotations are only available for 4862 images, and we have to further downsample the dataset to make it balanced with respect to disease ($Y$) and sex ($S$). Despite the size, we found that by starting with a pre-trained model we were able to get good convergence behaviour.
>
> In addition, we modified our setup to use the full 60k MNIST images, and, to better understand scaling behaviour with respect to dataset size, we compared the performance of the model trained on the full dataset to the one trained on the downsampled 6k variant. As expected, models trained on the larger variant performed better on average, however trends across subgroups were very similar (Figs D2,3). We thus believe our results are not an artifact of the small dataset size on CheXPert (further supported by our observation of the same trends on CelebA and civil_comments).
>
> **5. Overclaimed theoretical section**
>
> We have reworded the appropriate sections (please see reply 1 to Reviewer mP4g).
>
> **6. Choice of distance metric**
>
> After further consideration we changed the metric to KL divergence. We still see a strong correlation between the divergence of the weighted biased distribution to the target distribution and unbiased generalisation (Fig 5). We note that KL divergence and MAE produce quite similar results, in particular in subgroup ordering (Tab 2).
>
> **7. Limited novelty of Fig 3 in comparison to [2]**
>
> We acknowledge those experiments share similarities with [2], and we appreciate how their findings align with ours, reinforcing both conclusions. We now mention [2] in our manuscript. However, our evaluations differ in scope, as we consider a broader set of methods, including GroupDRO, CFair, and DomainInd, and focus on full fine-tuning rather than last-layer retraining.
>
> **8. Discuss references which study definitions of subgroups across spurious features**
>
> We appreciate this suggestion and have added a discussion about this type of study in our related work. Their findings complement ours by exploring how subgroup choice impacts mitigation in a setting where there are multiple SCs, while we focus on a simpler setting where there is only one SC but multiple additional variables (e.g. possible mitigation targets).
>
> **9. Minor issue with the paper’s assertion l.121**
>
> We have adjusted our statement accordingly and incorporated more citations. We further elaborate in response 2 to Reviewer 86pJ.
>
> **10. Minor edits**
>
> We thank the reviewer for noting these and have edited the manuscript.
>
> [1]Idrissi et al, CLeaR (2022)
>
> [2]Stromberg et al, TMLR (2024)

---

> > ### Comment · Reviewer_6xGC · 2025-04-02
> >
> > Thank you for the very detailed response. I greatly appreciate the substantial effort you put into running 200+ experiments in a short rebuttal timeframe. I found the additional experiments interesting, especially the CivilComments/CelebA and downsampling experiments, and their inclusion significantly improves the submission.
> >
> > I apologize for the lack of clarity on where the granular/coarse versions of CivilComments may be found in the literature. The version with four groups is used by, e.g., [1, 4], while the version with many groups is used by, e.g., [3, 5]. (They are the same dataset, from the WILDS benchmark [2], but one version collapses the identity categories). Nevertheless, your version is also a nice experiment.
> >
> > Based on the new empirical evidence, I have raised my score. I am prevented from raising it higher due to my outstanding concerns about the theoretical claims (though I appreciate the rewording of the theoretical overclaims and additional KL divergence experiments). Specifically, as mentioned in my review, the results are weakened by the lack of theoretical connection between the proposed metric and model performance, and the choice of distance (i.e., MAE vs KL) is insufficiently justified.
> >
> > [1] Kirichenko et al. Last layer re-training is sufficient for robustness to spurious correlations. ICLR 2023.
> >
> > [2] Koh et al. WILDS: A benchmark of in-the-wild distribution shifts. ICML 2021.
> >
> > [3] Liu et al. Just train twice: Improving group robustness without training group information. ICML 2021.
> >
> > [4] Sagawa et al. Distributionally robust neural networks for group shifts: On the importance of regularization for worst-case generalization. ICLR 2020.
> >
> > [5] Zhang et al. Correct-n-contrast: A contrastive approach for improving robustness to spurious correlations. ICML 2022.

---

> > > ### Author Response · Authors · 2025-04-04
> > >
> > > We thank the reviewer for raising their score and for recognising that the extra experiments have significantly improved the submission. We apologise for not further elaborating on the theoretical sections and our choice of distance metric in the initial rebuttal, this was due to the strict character limit for the rebuttal.
> > >
> > > Below, we provide additional clarifications on our choice of KL divergence along with some intuition on the theoretical connections between our metric and model performance. In light of these additional explanations, we hope the reviewer would consider further raising their score.
> > >
> > > **1. Choice of distance metric**
> > >
> > > We eventually chose the KL divergence to quantify the difference between the weighted biased training distributions and the target test distribution because it is more consistent with the literature. We noted similar uses of the KL divergence to compare train and test distributions in many recent works such as [1,2,3,5,6]. This allows our findings to be more easily compared to and extended to other settings, in contrast to our initial metric, MAE. We calculate it by comparing train and test discrete probability distributions of the 8 events corresponding to sampling each $(Y,A,S)$ combination.
> > >
> > > **2. Theoretical connection between proposed metric and model performance**
> > >
> > > While we agree that a complete proof explaining why the divergence between the weighted training distribution and the test distribution is correlated with test performance would be nice to have, we think this is out of scope for our current work. However, we would encourage future research to further explore this direction. In the meantime, **we provide additional intuition for this connection by framing our findings within established results in generalisation**.
> > >
> > > There is a broad consensus that “train/test distribution matching” through methods like data balancing can improve test set generalisation when the train and test distributions are not independent and identically distributed [4,7,9]. For example, a well known result by Ben David et al. [8] shows that for a model $h$ where the labelling function is the same across both source and target distributions:
> > > $$\mathcal{L_{test}}(h) \le \mathcal{L_{train}}(h) + D(\mathcal{P_{train}},\mathcal{P_{test}})$$
> > > where $D(\cdot,\cdot)$ represents the total variation divergence (TV).
> > >
> > > In our case, Pinsker’s inequality gives us $TV(\mathcal{P},\mathcal{Q}) \le \sqrt{\frac{1}{2}*KL(\mathcal{P} \parallel \mathcal{Q})}$. Since our labelling function does not change across distributions, we can show that the test error is upper bounded by:
> > >
> > > $$\mathcal{L_{unbiased}}(h) \le \mathcal{L_{train}}(h) + \sqrt{0.5*KL(\mathcal{P_{train}} \parallel \mathcal{P_{unbiased}})}$$
> > > We also note that in practice, since all our models reach a similar train error close to 0, the differences in upper bound are largely driven by the divergence between both distributions. In our setting we assume that the divergence between both distributions is attributable to differences in probabilities of sampling each $(Y,A,S)$ subgroup. This aligns with our findings showing that by reweighting the training distribution (through gDRO or resampling) and thus reducing the divergence to the test distribution, a lower generalisation error can be achieved.
> > >
> > > We further note other papers which similarly give expected generalisation error upper [2,3,5,6] and lower bounds [3] involving the KL divergence between training and test distributions. However, directly estimating generalisation under distribution shift is a difficult task which requires very strong assumptions [10].
> > >
> > > We believe that framing our findings within this theoretical line of work gives important context and further supports our observation that decreased KL divergence between the weighted training distribution and the test distribution results in improved unbiased generalisation. **We have incorporated this discussion into Section 5.3 of our paper.**
> > >
> > >
> > > [1] He et al., Information theoretic generalisation bounds for DNNs. InfoCog @ NeurIPS (2024)
> > >
> > > [2] Aminian et al., Learning algorithm generalization error bounds via auxiliary distributions. IEEE (2024).
> > >
> > > [3] Masiha et al., Learning under distribution mismatch and model misspecification. ISIT (2021).
> > >
> > > [4] Mansour et al., Domain adaptation: learning bounds and algorithms. COLT (2009).
> > >
> > > [5] Wu et al., On the Generalization for Transfer Learning: An Information-Theoretic Analysis. IEEE (2024).
> > >
> > > [6] Nguyen et al., KL guided domain adaptation. ICLR (2022).
> > >
> > > [7] Dong et al., How Does Distribution Matching Help Domain Generalization: An Information-theoretic Analysis. IEEE (2024).
> > >
> > > [8] Ben-David et al., A theory of learning from different domains. Machine Learning (2010).
> > >
> > > [9] Wang et al., Causal balancing for domain generalization. ICLR (2023).
> > >
> > > [10] Estimating generalization under distribution shifts via domain-invariant representations. ICML (2020).

---

### Official Review · Reviewer_mP4g · 2025-03-05

**Overall Recommendation:** 4

**Summary:**

The authors investigate the impact of group definitions on the performance of bias mitigation methods using semi-synthetic experiments on binary classification of images. Specifically, the authors introduce a spurious correlation into the training datasets by selecting examples based on two attributes and a label and then apply subgroup-based bias mitigation methods (gDRO, resampling, DomainInd, and CFair) using different subgroup definitions. Their evaluation reveals that subgroup definitions play a crucial role in model performance: the subgroup should capture the spurious correlation in the data to perform well. The authors quantify this finding using the minimum distance metric evaluated between the train dataset divided by subgroups and the unbiased test dataset. This metric seems to correlate well with bias mitigation performance, potentially revealing theoretical mechanisms behind the impact of subgroup definitions.

## update after rebuttal

The authors addressed all my questions. I will keep my evaluation.

**Claims And Evidence:**

I mostly have comments on positioning and writing.
1. Observing the correlation between the minimum distance metric and performance does not constitute a theoretical analysis. I think this finding should be positioned as a potential mechanism or explanation.
2. I think the paper should more clearly state that it focuses on the worst group accuracy and does not consider different wide-spread fairness criteria (e.g., demographic parity or equalized odds) and methods that enforce fairness constraints.
3. Additionally, while the authors acknowledge that they focus on spurious correlations in Section 5.6, I believe the paper should also clearly state that it only analyzes one specific model of spurious correlation.

**Essential References Not Discussed:**

I do not have additional suggestions for the references.

**Experimental Designs Or Analyses:**

I think the chosen bias mitigation methods (gDRO, resampling, DomainInd, and CFair) are reasonable. Many more recent modifications to these methods exist. However, the analysis of more established versions may be suitable for the exploratory study.

**Methods And Evaluation Criteria:**

The datasets and models seem reasonable. However, evaluating models on the dataset with "reversed" spurious correlation could be interesting: it would show whether the models leaned generalizable correlations or simply balanced accuracy across groups.

**Other Comments Or Suggestions:**

I do not have any suggestions.

**Other Strengths And Weaknesses:**

The paper is well-written, the findings seem original.

**Questions For Authors:**

1. How do you explain the success of subgroups based on $A$ paired with model-based methods (given that these subgroups perform poorly with reweighting-based methods)?
2. Could you argue why the results for your model of spurious transfer to more sophisticated scenarios?

**Relation To Broader Scientific Literature:**

The paper investigates an important question for group-based bias mitigation methods. These methods are fairly popular in the literature on fairness, robustness, and OOD generalization.

**Theoretical Claims:**

I have briefly looked at calculations in Appendix A.6.

---

> ### Author Rebuttal · Authors · 2025-04-01
>
> We thank the reviewer for their insightful comments, and we are glad to hear they appreciate the paper and the originality of our findings. We respond to the comments raised in the review below, and refer to additional results, which are all presented in this [folder](https://drive.google.com/drive/folders/1N5Fpf8VK41awBIYireIoaAjQES_i8xtV?usp=sharing).
>
> **1. Observing the correlation between the minimum distance metric and performance should be positioned as a potential mechanism**
>
> We agree with the reviewer’s comment and have changed the wording of this in the relevant sections, for instance in the introduction: “We provide a potential explanation for the differences observed based on the KL divergence between the optimal group-weighted biased distribution and the unbiased test distribution.”
>
> **2. Evaluating models on the dataset with "reversed" spurious correlation** could be interesting
>
> We are grateful for this suggestion, and performed the suggested evaluation to further probe the generalisation abilities of our different models. We find that for most datasets, bias mitigation methods, and subgroups, the performance suffers a small decrease on the dataset where the SC is “reversed” compared to the balanced unbiased test dataset (average of $-0.01$), as shown in Tab D1. We also see very similar trends across subgroups to the unbiased test dataset, as shown in Fig D1.
>
> **3. How do you explain the success of subgroups based on $A$ paired with model-based methods** (given that these subgroups perform poorly with reweighting-based methods)?
>
> We thank the reviewer for this question and agree that it is an important point to discuss. We touched on this briefly in the main text (“For model- based methods, a similar pattern is evident, with the added requirement that the subgroups contain both positive and negative classes”), but agree that this point deserves a more thorough and clear explanation.
>
> We came to the conclusion that to “map” a subgrouping from a reweighting-based method to a model-based method, the $Y$ component should be removed (e.g. each subgroup should contain both positive and negative classes). This is because methods like DomainInd and CFair learn representations for each subgroup separately. DomainInd trains a separate classifier for each subgroup, so it would not make sense to train a separate classification head for positive and negative classes. Similarly, CFair seeks to align subgroup representations, so it would not make sense to align representations of one subgroup containing only positive images to another subgroup containing only negative images, as this would defeat the point of training a discriminative classifier. On the other hand, for data-reweighting based methods, including the $Y$ in the subgroups helps to balance the final reweighted dataset with respect to class, and therefore improves results, especially in our case where the spurious correlation involves the class $Y$.
>
> This explains why we find that the subgroups which work well for DomainInd and CFair (e.g. $A$) are just a merged version of the ones which work well for gDRO and resampling (e.g. $AY$). To the best of our knowledge, **no papers have explicitly discussed this distinction despite its practical importance**. We have now elaborated on this in the main body of the paper and included a full discussion in the Appendix.
>
> **4. The paper should more clearly state that it focuses on worst group accuracy** and does not consider different wide-spread fairness criteria and methods that enforce fairness constraints
>
> We agree that this is an important point and have modified the manuscript to state this more clearly. In addition to the original sentence in the problem setup section “we frame the task according to the fairness paradigm described in Jones et al. (2025), whereby the objective is to generalise from a biased training distribution to an unbiased testing distribution”, we have added an extra sentence in the experimental setup to explicitly state that we are evaluating models based on overall performance on the unbiased dataset and worst-group performance. We selected these measures for their directness and simplicity compared to other fairness criteria.
>
> **5. The paper should clearly state that it only analyzes one specific model of spurious correlation**
>
> We have clarified this in the problem setup section.
>
> **6. Could you argue why the results for your model of spurious transfer to more sophisticated scenarios?**
>
> We believe the reviewer is asking if our results transfer to more complex datasets/tasks. We think that in more sophisticated scenarios there may be more complex SCs and causes for bias, so trends may vary on *which* specific subgroup is best, but we think our key point, that **subgroup definition impacts mitigation effectiveness**, holds across scenarios.
>
>
> We hope we have addressed the reviewer’s questions, but please let us know if there are additional points we could further clarify.

---

> > ### Comment · Reviewer_mP4g · 2025-04-02
> >
> > Thanks for the response!
> >
> > I will keep my score due to my internal evaluation of the significance of the findings.

---

> > > ### Author Response · Authors · 2025-04-04
> > >
> > > We thank the reviewer for their quick reply!

---

### Official Review · Reviewer_86pJ · 2025-03-12

**Overall Recommendation:** 3

**Summary:**

The paper investigates how subgroup definition impacts the effectiveness of bias mitigation methods in machine learning, hypothesizing that inconsistent success stems from this often-overlooked factor. Through experiments on semi-synthetic image classification tasks with varied subgroup definitions (coarse, fine-grained, intersectional, and noisy), the authors show that subgroup choice significantly influences outcomes, sometimes worsening fairness. Theoretical analysis reveals that the best subgrouping for bias mitigation is not always the one directly aligned with fairness objectives. Key contributions include introducing a novel setting with spurious correlations, demonstrating subgroup-dependent performance patterns, and providing theoretical insights into optimal grouping strategies, challenging conventional fairness assumptions.

## update after rebuttal

I believe the authors have put significant effort into the rebuttal, providing many new figures and tables in their shared folder. If these are integrated into the final camera-ready version, I believe the paper will be above the acceptance threshold and will make a strong contribution with meaningful results to the field. Therefore, I have raised my score accordingly.

**Claims And Evidence:**

1. **Overlap between first and fourth key contributions**: The claim that subgroup choice impacts disparities (first contribution) appears closely related to the finding that the best way to achieve fairness for a subgroup is not necessarily using it in bias mitigation (fourth contribution). Since both findings seem to be discussed in Section 5.4, the distinction between them is unclear. The authors should clarify how these two contributions differ or consolidate them if they are essentially the same.
2. **Empirical support for optimal grouping strategies (second contribution)**: While the authors suggest an optimal grouping strategy (A, Y, S), their evidence is primarily based on Colored MNIST, a synthetic dataset. This raises concerns about generalizability to real-world settings. More experiments on diverse datasets would strengthen the claim.
3. **Theoretical explanation (third contribution)**: The theoretical analysis is currently relegated to the appendix without clear integration into the main text. A more structured discussion in the main sections, with problem formulation and intuition, would improve clarity and support the claim more convincingly. Better section organization in A.5. and A.6. would also make the theoretical insights more accessible.

Overall, while the findings are interesting, clarifying overlapping claims, expanding empirical validation, and improving theoretical discussion would strengthen the paper’s contributions.

**Essential References Not Discussed:**

Yes, several relevant works are missing that would provide important context for the paper’s key contributions:
1. Whac-a-Mole (Sagawa et al.) – This work examines spurious correlations and the unintended consequences of bias mitigation, particularly in cases where debiasing one issue exacerbates another. It directly relates to the paper’s argument that subgroup choice can lead to worse outcomes.
2. DFR (Kirichenko et al.), AFR (Qiu et al.), EVaLS (Ghaznavi et al.), and SELF (LaBonte et al.) – These methods focus on fairness interventions that do not rely on explicit subgroup labels for debiasing or model selection. Since the paper critiques subgroup definition as a bottleneck, these works provide alternative solutions that should be acknowledged.

Citing and discussing these works would provide a more comprehensive view of the challenges in bias mitigation and situate the paper’s findings within the broader literature.

**Experimental Designs Or Analyses:**

Yes, the experimental design and analyses are generally valid. The subgroup generation strategy is comprehensive, and the evaluation setup is well-structured. Theoretical insights align with empirical findings, supporting the study’s conclusions. However, adding key baselines and alternative metrics like Worst-Group Accuracy (WGA) would further enhance validity.

**Methods And Evaluation Criteria:**

The proposed evaluation framework is well-structured and relevant for analyzing the impact of subgroup choice on bias mitigation. The subgroup generation strategy is comprehensive, covering various realistic scenarios, including noisy annotations, coarse vs. fine-grained groupings, and intersectional subgroups. The use of Colored MNIST (synthetic) and CXP (real-world chest X-ray dataset) ensures both controlled and practical evaluations. However, some areas could be improved:
1. Evaluation Metrics: While AUC is a valid metric, it is not widely used in this literature. Including Worst-Group Accuracy (WGA) would provide better insights, making results more comparable to prior work.
2. Baseline Comparisons: The paper lacks comparisons with key existing methods, such as Whac-a-Mole (which accounts for both known and unknown spurious correlations) and approaches like DFR (Kirichenko et al.), AFR (Qiu et al.), EVaLS (Ghaznavi et al.), and SELF (LaBonte et al.). Many of these do not rely on group labels for debiasing or model selection, making them highly relevant benchmarks. Their inclusion would clarify the relative effectiveness of subgroup selection in bias mitigation.
3. Generalizability & Real-World Validation: While the paper includes a real-world dataset (CXP), additional benchmarks from other domains would strengthen the findings. This is particularly important given the reliance on Colored MNIST for some subgrouping insights, which may not always translate to real-world settings.

Overall, the methodology is well-motivated, but incorporating WGA, broader baselines, and additional real-world benchmarks would enhance the study’s impact and comparability.

**Other Comments Or Suggestions:**

L222 “drops the on unbiased test set” → “drops on the unbiased test set”

**Other Strengths And Weaknesses:**

One of the key strengths of the paper is its novel approach to identifying optimal subgroups for bias mitigation, as well as its thorough benchmarking across various subgroup combinations. This is a meaningful contribution that can help guide future work on improving fairness in machine learning models. The systematic evaluation and detailed subgroup generation strategy are well-executed, providing valuable insights into how subgroup choices can affect model performance.

However, the paper could benefit from more evidence or case studies demonstrating how this optimal subgroup identification can be applied in a real-world pipeline. While the theoretical analysis and experimental results are strong, additional practical examples or a clearer link to real-world applications would further solidify the significance and applicability of the approach.

**Questions For Authors:**

1. Clarity of First Key Contribution: Could you please clarify the distinction between the first and fourth key contributions?
2. Empirical Results for Optimal Grouping Strategies: Can you provide more empirical results or case studies demonstrating how the optimal subgroup strategies perform in real-world datasets, beyond the synthetic or semi-synthetic examples (e.g., coloured MNIST)?
3. Theoretical Insights: The theoretical discussion in the appendix feels somewhat disconnected from the main text. Could you summarize the key theoretical insights more explicitly in the main body of the paper to help readers better understand the underlying mechanics of your findings?

**Relation To Broader Scientific Literature:**

The paper builds on existing work in fairness and bias mitigation by challenging the assumption that the best way to improve fairness for a specific subgroup is to use that subgroup in mitigation. This aligns with prior findings in Whac-a-Mole (Sagawa et al.), which showed that bias mitigation can shift bias rather than eliminate it. It also relates to DFR (Kirichenko et al.), AFR (Qiu et al.), and SELF (LaBonte et al.), which explore debiasing without explicit group labels, suggesting alternative ways to approach fairness. Additionally, the paper’s theoretical analysis on subgroup weighting connects to prior work like EVaLS (Ghaznavi et al.), which examines model selection in biased settings. By emphasizing subgroup choice, the paper provides a novel perspective on why bias mitigation often fails, complementing and extending these prior studies.

**Theoretical Claims:**

Yes I have checked the theoretical claims and proofs but it was not so easy to follow and understand for me.

---

> ### Author Rebuttal · Authors · 2025-04-01
>
> We thank the reviewer for their detailed review and helpful suggestions. We summarise the >200 supplementary experiments and edits we have made to address them below. All extra results are shared in this [folder](https://rb.gy/xebqmp). We hope this will give the reviewer confidence to raise their score.
>
> **1. More experiments on diverse datasets** and additional practical examples or a clearer link to real-world applications would further solidify the significance
>
> We agree with the reviewer's suggestion and have **expanded our analysis to two more datasets** with different modalities and complexity: civil_comments (text) and CelebA (natural images). We find that results closely align across the four datasets. They show similar patterns on the impact of subgroups on mitigation effectiveness with performance strongly correlating to the ability to restore the unbiased distribution. Please see details on the updated setup in Tab A3 and key results in Figs 2 and 5. We believe this strengthens the papers’ claims.
>
> We have clarified the link to real-world applications by further discussing practical examples like differences in the proportion of chest drains causing sex disparities and by including real results on the effect of granularity of subgroups in civil_comments as discussed in response 9 to Reviewer 6xGC.
>
> **2. Lacks comparisons** with key existing methods, many of these do not rely on group labels
>
> We thank the reviewer for their comment and relevant references. We have added them to our related work section, and have in particular expanded our discussion on methods which do not require subgroup labels as they do provide important context for our work.
>
> For our experiments, we focus on a select few established and simple methods as a foundation since many newer approaches (like DFR or AFR) share core principles with resampling and GroupDRO. A recent benchmark also showed no significant performance differences between mitigation techniques [1].
>
> We initially restricted our experiments to methods with subgroup labels because (a) it is important to understand how subgroups affect mitigation as even methods which do not rely on labels often infer these subgroups (as Reviewer 6xGC noted), and (b) many methods which do not require subgroup labels often actually require some labels on the validation data (e.g. for hyperparameter selection), as discussed in [2], and hence subgroup definition remains an important question, and finally (c), methods with labels are often the upper bound [3-5], though recent exceptions exist [6,7].
>
> However, we do agree they could provide an interesting comparison, and so conducted additional experiments on Just Train Twice (JTT) [8], which does not use subgroup labels for training (but requires some for model selection). As shown in Fig C1 and Tab C1, we find that with validation subgroup labels to guide model and hyperparameter selection JTT performs mostly on par with our other methods, however, **performance is again highly dependent on the choice of subgroups**. When no subgroup annotations are used (i.e. model selection is done by overall validation accuracy), the method does not improve over ERM (except for on MNIST where JTT works remarkably effectively, most likely due to the simplicity of the task). We have devoted a new section in our Appendix to these results.
>
> **3. Overlap between first and fourth key contributions**
>
> The second contribution demonstrates broadly how subgroup choice impacts mitigation effectiveness, while the fourth contribution highlights a specific counterintuitive finding: that achieving fairness between two particular groups may require using different groups for mitigation rather than those groups themselves. We've clarified this distinction in the manuscript, as we believe this specific insight represents an important and non-obvious result.
>
> **4. Structure of theoretical section**
>
> We thank the reviewer for their comment and have modified the manuscript to improve the clarity of this section by including key intuition, examples, and restructuring A5 and A6. We believe this should significantly help readers better understand the mechanics of our findings.
>
> **5. While AUC is a valid metric, it is not widely used in this literature**. Including WGA would provide better insights
>
> We include WGA in Tables 2 and 4. For the other results, we use overall AUC for consistency with [1]. Moreover, since we frame the objective as generalisation to an unbiased test set (as in [9]), overall AUC is a useful measure of generalisation performance. AUC is also threshold-independent, which helps give confidence that our results are not simply a side-effect of poor/biased threshold selection.
>
> [1]Zong et al, ICLR (2023)
>
> [2]Pezeshki et al, ICML (2024)
>
> [3]Taghanaki et al, NeurIPS (2022)
>
> [4]Bayasi et al, MICCAI (2024)
>
> [5]Han et al, ICML (2024)
>
> [6]Liu et al, ICLR (2023)
>
> [7]Pezeshki et al, NeurIPS (2021)
>
> [8]Liu et al, ICML (2021)
>
> [9]Jones et al, ICLR (2025)

---

> > ### Comment · Reviewer_86pJ · 2025-04-02
> >
> > I would like to thank and congratulate authors for their amazing efforts on the rebuttal. I have adjusted my score accordingly. All the best.

---

> > > ### Author Response · Authors · 2025-04-04
> > >
> > > We thank the reviewer for their very prompt reply and are happy to hear we have addressed their concerns!

---

### Decision · Program_Chairs · 2025-05-01

**Decision:**

Accept (poster)

**Comment:**

The paper empirically shows that bias mitigation methods with a given set of subgroups don't necessarily work for other subgroups not included for the mitigation. This finding is not surprising because these mitigation approaches specifically aim to achieve fairness with respect to the given subgroups. It is not possible for any methods to guarantee that there will be no performance gap between any unknown arbitrary subgroups. Nevertheless, the reviewers felt that it is still worthwhile reporting this empirical result.

Another critical issue of the paper was evaluation. The reviewers had concerns about insufficient evaluation even though the paper is purely empirical. The authors added a large amount of evaluation results from new experiments. These results should have been conducted and included in the original paper submission such that the reviewers can have full access to the content. Furthermore, it would be unfair to other authors who submitted (or couldn't submit) all the evidence before the submission due.

Despite these concerns, AC acknowledges the unanimous support from the reviewers. If this paper is accepted, the authors should incorporate all the suggestions from the reviewers and the new results in the paper.